# ROBUST DETERMINANTAL GENERATIVE CLASSIFIER FOR NOISY LABELS AND ADVERSARIAL ATTACKS

## ABSTRACT

Large-scale datasets may contain significant proportions of noisy (incorrect) class labels, and it is well-known that modern deep neural networks poorly generalize from such noisy training datasets. In this paper, we propose a novel inference method, *Deep Determinantal Generative Classifier (DDGC)*, which can obtain a more robust decision boundary under any softmax neural classifier pre-trained on noisy datasets. Our main idea is inducing a generative classifier on top of hidden feature spaces of the discriminative deep model. By estimating the parameters of generative classifier using the minimum covariance determinant estimator, we significantly improve the classification accuracy, with neither re-training of the deep model nor changing its architectures. In particular, we show that DDGC not only generalizes well from noisy labels, but also is robust against adversarial perturbations due to its large margin property. Finally, we propose the ensemble version of DDGC to improve its performance, by investigating the layer-wise characteristics of generative classifier. Our extensive experimental results demonstrate the superiority of DDGC given different learning models optimized by various training techniques to handle noisy labels or adversarial samples. For instance, on CIFAR-10 dataset containing 45% noisy training labels, we improve the test accuracy of a deep model optimized by the state-of-the-art noise-handling training method from 33.34% to 43.02%.

## 1 INTRODUCTION

Deep neural networks (DNNs) are known to generalize well when they are trained on large-scale datasets with clean label annotations. For example, DNNs have achieved state-of-the-art performance on many classification tasks, e.g., speech recognition (Amodei et al., 2016), object detection (Girshick, 2015) and image classification (He et al., 2016). However, as the scale of the training dataset increases, it becomes infeasible to obtain all class labels from domain experts. A common practice is collecting the class labels from data mining on social media (Mahajan et al., 2018) and web data (Krause et al., 2016). However, they may contain missing/noisy (incorrect) labels, and recent studies have shown that modern deep architectures may generalize poorly from the noisy datasets (Zhang et al., 2017; Arpit et al., 2017). For example, in Figure 1(a), the test set accuracy (black line) of DenseNet-100 model (Huang & Liu, 2017) trained on the CIFAR-10 dataset (Krizhevsky & Hinton, 2009) significantly decreases as the noise fraction increases.

To overcome the poor generalization issue of DNNs against noisy labels, many training strategies have been investigated in the literature. Reed et al. (2014) proposed a bootstrapping method which trains deep models with new labels generated by a convex combination of the raw (noisy) labels and their predictions, and Ma et al. (2018b) improved the bootstrapping method by utilizing the dimensionality of subspaces during training. Patrini et al. (2017) modified the loss and posterior distribution to eliminate the influence of noisy labels, and Hendrycks et al. (2018) improved such a loss correction method by utilizing the information from data with true class labels. Goldberger & Ben-Reuven (2017) added an additional softmax layer to model a noise transition matrix. Finally, training DNNs on selected samples also has been studied (Jiang et al., 2018; Ren et al., 2018; Malach & Shalev-Shwartz, 2017; Han et al., 2018). However, adopting such training methods might incur expensive back-and-forth costs, e.g., additional time and hyperparameter tuning. This motivates our approach of developing a more plausible inference method which can be applied to any pre-trained

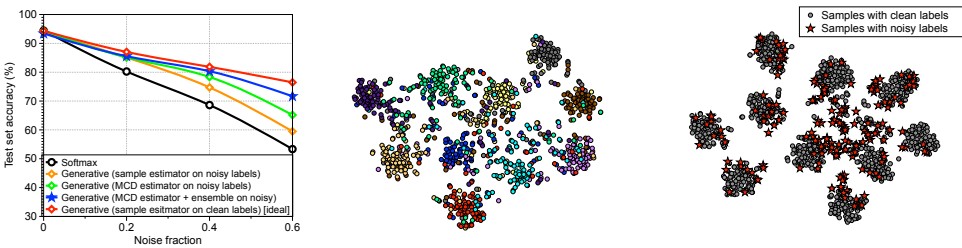

(a) Test set accuracy comparison by varying noise fraction

(b) Features on penultimate layer from test samples by t-SNE

(c) Features on penultimate layer from training samples by t-SNE

Figure 1: Experimental results under DenseNet-100 model and CIFAR-10 dataset. (a) Test accuracy comparison when the labels of a given proportion of training samples are flipped to other labels uniformly at random. Visualization of features on the penultimate layer using t-SNE from (b) test samples (same colors indicate same classes) and (c) training samples when the noise fraction is 20%.

deep model. Nevertheless, our direction is complementary to the prior works, where one can also combine them for even better performance (see Table 3 and 4 in Section 3).

**Contributions.** It has been observed that DNNs can learn meaningful feature patterns shared by multiple training examples even for datasets with noisy labels (Arpit et al., 2017). We also found that an induced generative classifier (Ng & Jordan, 2002) under linear discriminant analysis (LDA) assumption (with naive estimations on sample mean and covariance) built upon the hidden feature space can outperform the softmax classifier (orange line in Figure 1(a)). In Figure 1(b), the hidden features from test samples projected in a 2-dimensional space using t-SNE (Maaten & Hinton, 2008) are illustrated. Here, one can observe that all ten classes are well-separated in the embedding space, even though the model is trained under a noisy dataset. More importantly, Figure 1(c) plotting noisy training samples implies that outliers induce the class-wise multi-modal distributions in the feature space. Therefore, an LDA-like generative classifier assuming the class-wise unimodal distribution might be more robust, as a discriminative classifier is easier to be overfitted by outliers. This motivates our goal to induce a generative classifier on the pre-trained hidden features of DNNs.

To this end, we propose the so-called deep determinantal generative classifier (DDGC), based on the minimum covariance determinant (MCD) (Rousseeuw, 1984; Rousseeuw & Driessen, 1999) estimation on its parameters. While a naive sample estimator can be highly influenced by outliers, MCD estimator can improve the robustness by removing them. Note that MCD is known to have a near-optimal breakdown point (Hampel, 1971) of almost 50% in most situations, i.e., the number of outliers should be larger than that of normal samples to fool it. We further provide a new theoretical support on the larger margin property of DDGC such that not only does it generalize well from noisy labels (Durrant & Kabán, 2010), but also improves the robustness against adversarial perturbations (Pang et al., 2018). In addition, we observe that DNNs tend to have similar hidden features, regardless of whether they are trained with clean or noisy labels at early layers (Morcos et al., 2018), and DDGC built from low-level features can be often more effective. Under the observation, we finally propose an ensemble version of DDGC to incorporate all effects of low and high layers.

We demonstrate the effectiveness of DDGC using modern neural architectures, such as DenseNet (Huang & Liu, 2017) and ResNet (He et al., 2016) trained for image classification tasks including CIFAR (Krizhevsky & Hinton, 2009) and SVHN (Netzer et al., 2011). First, our methods (green and blue lines in Figure 1(a)) significantly outperform the softmax classifier, although they use the same feature representations trained by the noisy dataset. For example, we improve the test accuracy of DenseNet on CIFAR-10 datasets with 60% noisy labels from 53.34% to 74.72%. We also demonstrate that DDGC can be used to further improve various prior training methods (Reed et al., 2014; Patrini et al., 2017; Ma et al., 2018b; Han et al., 2018; Jiang et al., 2018; Malach & Shalev-Shwartz, 2017) which are specialized to handle the noisy environment. Finally, DDGC is shown to be robust against various adversarial attacks (Goodfellow et al., 2015; Moosavi Dezfooli et al., 2016; Carlini & Wagner, 2017).

## 2 ROBUST INFERENCE VIA GENERATIVE CLASSIFIER

In this section, we propose a novel inference method which obtains a robust posterior distribution from any softmax neural classifier pre-trained on datasets with noisy labels. Our idea is inducing the generative classifier using fixed features from any pre-trained model. We show the advantages of our method in terms of high breakdown points (Hampel, 1971), generalization error (Durrant & Kabán, 2010) and adversarial robustness (Pang et al., 2018). We also investigate the layer-wise characteristics of generative classifiers, and introduce an ensemble of them to improve its performance. We emphasize again that we study inference methods, not requiring any modification on pre-trained deep models. Hence, the proposed methods are easily applicable on top of any pre-trained modern neural classifiers.

### 2.1 GENERATIVE CLASSIFIER AND MCD ESTIMATOR

Let $\mathbf{x}$ be an input and $y \in \{1, \cdots, C\}$ be its class label. Without loss of generality, suppose that a pre-trained softmax neural classifier is given: $P(y = c|\mathbf{x}) = \frac{\exp\left(\mathbf{w}_c^\top f(\mathbf{x}) + b_c\right)}{\sum_{c'} \exp\left(\mathbf{w}_{c'}^\top f(\mathbf{x}) + b_{c'}\right)}$, where $\mathbf{w}_c$ and $b_c$ are the weight and the bias of the softmax classifier for class $c$, and $f(\cdot) \in \mathbb{R}^d$ denotes the output of the penultimate layer of DNNs. Then, without any modification on the pre-trained softmax neural classifier, we induce a generative classifier by assuming the class-conditional distribution follows the multivariate Gaussian distribution. In particular, we define $C$ Gaussian distributions with a tied covariance $\boldsymbol{\Sigma}$, i.e., linear discriminant analysis (LDA) (Fisher, 1936), and a Bernoulli distribution for the class prior: $P(f(\mathbf{x})|y = c) = \mathcal{N}(f(\mathbf{x})|\mu_c, \boldsymbol{\Sigma})$, $P(y = c) = \beta_c$, where $\mu_c$ is the mean of multivariate Gaussian distribution and $\beta_c$ is the normalized prior for class $c$. We provide an analytic justification on the LDA (i.e., tied covariance) assumption in Appendix A. Then, based on the Bayesian rule, we induce a new posterior different from the softmax one as follows:

$$P(y = c|f(\mathbf{x})) = \frac{P(y = c)P(f(\mathbf{x})|y = c)}{\sum_{c'} P(y = c')P(f(\mathbf{x})|y = c')} = \frac{\exp\left(\mu_c^\top \boldsymbol{\Sigma}^{-1} f(\mathbf{x}) - \frac{1}{2}\mu_c^\top \boldsymbol{\Sigma}^{-1}\mu_c + \log \beta_c\right)}{\sum_{c'} \exp\left(\mu_{c'}^\top \boldsymbol{\Sigma}^{-1} f(\mathbf{x}) - \frac{1}{2}\mu_{c'}^\top \boldsymbol{\Sigma}^{-1}\mu_{c'} + \log \beta_{c'}\right)}.$$

To estimate the parameters of the generative classifier, one can compute the sample class mean and covariance of training samples $\mathcal{X}_N = \{(\mathbf{x}_1, y_1), \ldots, (\mathbf{x}_N, y_N)\}$:

$$\bar{\mu}_c = \sum_{i:y_i=c} \frac{f(\mathbf{x}_i)}{N_c}, \quad \bar{\boldsymbol{\Sigma}} = \sum_c \sum_{i:y_i=c} \frac{(f(\mathbf{x}_i) - \bar{\mu}_c)(f(\mathbf{x}_i) - \bar{\mu}_c)^\top}{N}, \quad \bar{\beta}_c = \frac{N_c}{N}, \quad (1)$$

where $N_c$ is the number of training samples labeled to be class $c$. We remark that inducing a generative classifier (e.g., a mixture of Gaussian) on pre-trained deep models was studied for various purposes, e.g., speech recognition (Hermansky et al., 2000) and novelty detection (Lee et al., 2018). However, the setting of noisy training labels was not investigated under existing approaches on this line. To handle this, we design a more advanced generative classifier as stated in below.

One can expect that the naive sample estimator (1) can be highly influenced by outliers (i.e., training samples with noisy labels). In order to improve the robustness, we propose the so-called deep determinantal generative classifier (DDGC), which utilizes the minimum covariance determinant (MCD) estimator (Rousseeuw & Driessen, 1999) to estimate its parameters. For each class $c$, the main idea of MCD is finding a subset $\mathcal{X}_{K_c}$ for which the determinant of the corresponding sample covariance is minimized:

$$\min_{\mathcal{X}_{K_c} \subset \mathcal{X}_{N_c}} \det\left(\widehat{\boldsymbol{\Sigma}}_c\right) \quad \text{subject to } |\mathcal{X}_{K_c}| = K_c, \quad (2)$$

where $\mathcal{X}_{N_c}$ is the set of training samples labeled to be class $c$, $\widehat{\boldsymbol{\Sigma}}_c$ is the sample covariance of $\mathcal{X}_{K_c}$ and $0 < K_c < N_c$ is a hyperparameter. Then, only using the samples in $\bigcup_c \mathcal{X}_{K_c}$, it estimates the parameters, i.e., $\widehat{\mu}_c, \widehat{\boldsymbol{\Sigma}}, \widehat{\beta}_c$, of the generative classifier, by following (1). Such a new estimator can be more robust by removing the outliers which might be widely scattered in datasets. The robustness of MCD estimator has been justified in the literature: it is known to have near-optimal breakdown points (Hampel, 1971), i.e., the smallest fraction of data points that need to be replaced by arbitrary values (i.e., outliers) to fool the estimator completely. Formally, denote $\mathcal{Y}_M$ as a set obtained by replacing $M$ data points of set $\mathcal{Y}$ by some arbitrary values. Then, for a multivariate mean estimator

$\mu = \mu(\mathcal{Y})$ from $\mathcal{Y}$, the breakdown point is defined as follows (see Appendix A for more detailed explanations including the breakdown point of covariance estimator):

$$\varepsilon^*(\mu, \mathcal{Y}) = \frac{1}{|\mathcal{Y}|} \min \left\{ M \in \{1, \cdots, |\mathcal{Y}|\} : \sup_{\mathcal{Y}_M} \|\mu(\mathcal{Y}) - \mu(\mathcal{Y}_M)\| = \infty \right\}.$$

While the breakdown point of the naive sample estimator is 0%, the MCD estimator for the generative classifier under LDA assumption is known to attain its breakdown value of $\min_c \frac{\lfloor (N_c - d + 1)/2 \rfloor}{N_c} \approx 50\%$ (Lopuhaä et al., 1991). Inspired by this fact, we choose the default value of $K_c$ in (2) by $\lfloor (N_c + d + 1)/2 \rfloor$.

We also establish the following theoretical support that the MCD-based generative classifier, i.e., DDGC, can have smaller errors on parameter estimations and produce a larger margin, compared to the naive sample estimator, under some assumptions for its analytic tractability.

**Theorem 1.** *Assume the followings:*

(A1) *The distribution of hidden features is $P(f(\mathbf{x})|y = c) = \mathcal{N}\left(f(\mathbf{x})|\mu_c, \sigma^2\mathbf{I}\right)$ (i.e., the conditional Gaussian distribution) and that of outliers is an arbitrary distribution with mean $\mu_{\text{out}}$ and covariance matrix $\sigma^2_{\text{out}}\mathbf{I}$, where $\mathbf{I} \in \mathbb{R}^{d \times d}$ is the identity matrix.*

(A2) *$\mu_{\text{out}} = \mathbf{0}$, and $\frac{1}{C}\sum_c \mu_c\mu_c^T$ is a diagonal matrix.*

(A3) *All classes have the same number of samples (i.e., $N_c = \frac{N}{C}$), the same fraction $\delta_{\text{out}} < 1$ of outliers, and the sample fraction $\delta_{\text{mcd}} = \frac{K_c}{N_c} < 1$ of samples selected by MCD estimator.*

(A4) *The outliers are widely scattered such that $\sigma^2 < \sigma^2_{\text{out}}$.*

(A5) *The number of outliers is not too large such that $\delta_{\text{out}} < 1 - \delta_{\text{mcd}}$ and $\delta_{\text{mcd}} > \frac{d}{N_c}$.*

*Let $\widehat{\mu}, \widehat{\boldsymbol{\Sigma}}$ and $\bar{\mu}, \bar{\boldsymbol{\Sigma}}$ be the outputs of the MCD and sample estimators, respectively. Then, $\widehat{\mu}, \bar{\mu}, \widehat{\boldsymbol{\Sigma}}, \bar{\boldsymbol{\Sigma}}$ converge almost surely to their expectation as $N \to \infty$, and it holds that*

$$\frac{B(\widehat{\mu}_c, \widehat{\mu}_{c'}, \boldsymbol{\Sigma}, \widehat{\boldsymbol{\Sigma}})}{B(\bar{\mu}_c, \bar{\mu}_{c'}, \boldsymbol{\Sigma}, \bar{\boldsymbol{\Sigma}})} \xrightarrow{\text{a.s.}} \lim_{N \to \infty} \frac{B(\widehat{\mu}_c, \widehat{\mu}_{c'}, \boldsymbol{\Sigma}, \widehat{\boldsymbol{\Sigma}})}{B(\bar{\mu}_c, \bar{\mu}_{c'}, \boldsymbol{\Sigma}, \bar{\boldsymbol{\Sigma}})} \leq 1, \tag{3}$$

$$\|\mu_c - \widehat{\mu}_c\|_1 \xrightarrow{\text{a.s.}} \lim_{N \to \infty} \|\mu_c - \widehat{\mu}_c\|_1 = 0, \qquad \|\mu_c - \bar{\mu}_c\|_1 \xrightarrow{\text{a.s.}} \lim_{N \to \infty} \|\mu_c - \bar{\mu}_c\|_1 = \delta_{\text{out}}\|\mu_c\|_1 \tag{4}$$

$$\frac{(\widehat{\mu}_c - \widehat{\mu}_{c'})^T \widehat{\boldsymbol{\Sigma}}^{-1}(\widehat{\mu}_c - \widehat{\mu}_{c'})}{(\bar{\mu}_c - \bar{\mu}_{c'})^T \bar{\boldsymbol{\Sigma}}^{-1}(\bar{\mu}_c - \bar{\mu}_{c'})} \xrightarrow{\text{a.s.}} \lim_{N \to \infty} \frac{(\widehat{\mu}_c - \widehat{\mu}_{c'})^T \widehat{\boldsymbol{\Sigma}}^{-1}(\widehat{\mu}_c - \widehat{\mu}_{c'})}{(\bar{\mu}_c - \bar{\mu}_{c'})^T \bar{\boldsymbol{\Sigma}}^{-1}(\bar{\mu}_c - \bar{\mu}_{c'})} = \frac{1}{(1 - \delta_{\text{out}})^2} \geq 1. \tag{5}$$

*for all $c$, $c'$, where $B(\widehat{\mu}_c, \widehat{\mu}_{c'}, \boldsymbol{\Sigma}, \widehat{\boldsymbol{\Sigma}}) := \exp\left(-\frac{1}{8}\frac{[(\widehat{\mu}_c - \widehat{\mu}_{c'})^T \widehat{\boldsymbol{\Sigma}}^{-1}(\widehat{\mu}_c - \widehat{\mu}_{c'})]^2}{(\widehat{\mu}_c - \widehat{\mu}_{c'})^T \widehat{\boldsymbol{\Sigma}}^{-1}\boldsymbol{\Sigma}\widehat{\boldsymbol{\Sigma}}^{-1}(\widehat{\mu}_c - \widehat{\mu}_{c'})}\right).$*

The proof of the above theorem is given in Appendix F, where it is built upon the fact that the determinants can be expressed as the $d$-th degree polynomial of outlier ratio. We note that one might enforce the assumptions of the diagonal covariance matrices in $\mathcal{A}1$ and the zero-mean/uncorrelated properties of class mean in $\mathcal{A}2$ to hold under an affine translation of hidden features. In addition, the assumption in $\mathcal{A}5$ holds when $N_c$ is large enough. Nevertheless, we think most assumptions of Theorem 1 are not necessary to claim the superiority of DDGC (they are rather from a limitation of our proof techniques) and it is an interesting future direction to explore to relax them.

First, (5) implies that the MCD estimator induces a larger margin and improves the robustness to adversarial attacks, since Pang et al. (2018) showed that the margin of a generative classifier corresponds to the Mahalanobis distance between class-conditional distributions and the optimal robustness to adversarial samples is achieved by the maximum margin. More importantly, (3) and (4) together imply that a MCD-based classifier can generalize better since the generalization error of a generative classifier is known to be bounded as follows (Durrant & Kabán, 2010):

$$P_{\mathbf{x}}\left(y^* \neq \arg\max_y P_{\widehat{\mu}_c, \widehat{\boldsymbol{\Sigma}}}(y|\mathbf{x})\right) \leq \sum_c \sum_{c' \neq c} B(\widehat{\mu}_{c'}, \widehat{\mu}_c, \boldsymbol{\Sigma}, \widehat{\boldsymbol{\Sigma}}) + D\sum_c \|\mu_c - \widehat{\mu}_c\|_1,$$

for some constant $D > 0$.

---

**Algorithm 1** (Rousseeuw & Driessen, 1999) Approximating MCD for a single Gaussian.

**Input:** $\mathcal{X}_{N_c} = \{\mathbf{x}_i : i = 1, \cdots, N_c\}$ and the maximum number of iterations $I_{\max}$.

---

Uniformly sample initial subset $\mathcal{X}_{K_c} \subset \mathcal{X}_{N_c}$, where $|\mathcal{X}_{K_c}| = \lfloor (N_c + d + 1)/2 \rfloor$.
Compute $\widehat{\mu}_c = \frac{1}{|\mathcal{X}_{K_c}|} \sum_{\mathbf{x} \in \mathcal{X}_{K_c}} f(\mathbf{x})$, $\widehat{\boldsymbol{\Sigma}}_c = \frac{1}{|\mathcal{X}_{K_c}|} \sum_{\mathbf{x} \in \mathcal{X}_{K_c}} (f(\mathbf{x}) - \widehat{\mu}_c)(f(\mathbf{x}) - \widehat{\mu}_c)^\top$.

**for** $i = 1$ **to** $I_{\max}$ **do**
    Compute the Mahalanobis distance: $\alpha(\mathbf{x}) = (f(\mathbf{x}) - \widehat{\mu}_{\mathbf{c}})^\top \widehat{\boldsymbol{\Sigma}}_c^{-1} (f(\mathbf{x}) - \widehat{\mu}_{\mathbf{c}})$, $\forall \mathbf{x} \in \mathcal{X}_{N_c}$.
    Update $\mathcal{X}_{K_c}$ such that it includes $\lfloor (N_c + d + 1)/2 \rfloor$ samples with smallest distance $\alpha(\mathbf{x})$.
    Compute sample means and covariance, i.e., $\widehat{\mu}_c, \widehat{\boldsymbol{\Sigma}}_c$, using new subset $\mathcal{X}_{K_c}$.
    Exit the loop if the determinant of covariance matrix is not decreasing anymore.
**end for**
Return $\widehat{\mu}_c$ and $\widehat{\boldsymbol{\Sigma}}_c$

---

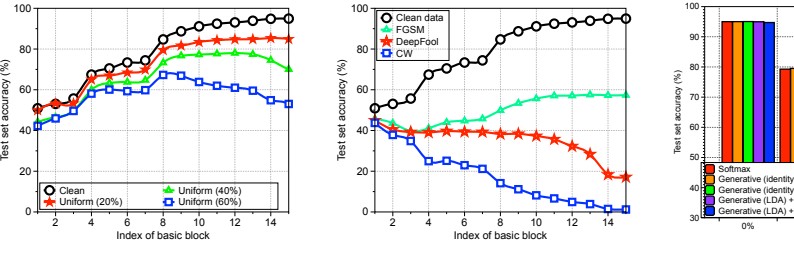

(a) Generalization of generative classifiers from noisy labels
(b) Robustness of generative classifiers to adversarial attacks
(c) Generative classifiers under various assumptions

Figure 2: Experimental results under ResNet-34 model and CIFAR-10 dataset. (a) Test accuracy of generative classifiers computed at different basic blocks. (b) Test accuracy on various adversarial attacks when the model is trained on clean dataset. (c) Test accuracy of generative classifiers from penultimate features under various assumptions: identity covariance and tied covariance (LDA).

## 2.2 APPROXIMATION ALGORITHM FOR MCD

Even though the MCD estimator has several advantages, the optimization (2) is computationally intractable (i.e., NP-hard) to solve (Bernholt, 2006). To handle this issue, we aim for computing its approximate solution, following a similar idea to that by (Hubert & Van Driessen, 2004). We design two step scheme as follows: (a) obtain the mean and covariance, i.e., $\widehat{\mu}_c, \widehat{\boldsymbol{\Sigma}}_c$, using Algorithm 1 for each class $c$, and (b) compute the tied covariance by $\widehat{\boldsymbol{\Sigma}} = \frac{\sum_c K_c \widehat{\boldsymbol{\Sigma}}_c}{\sum_c K_c}$. In other words, we apply the MCD estimator for each class, and combine the individual covariances into a single one due to the tied covariance assumption of LDA. Even though finding the optimal solution of MCD estimator under a single Gaussian distribution is still intractable, Algorithm 1 can produce a local optimal solution since it monotonically decreases the determinant under any random initial subset (Rousseeuw & Driessen, 1999). We choose $I_{\max} = 2$ in our experiments as the additional iterations would not improve the results significantly.

## 2.3 ENSEMBLE OF GENERATIVE CLASSIFIERS

To further improve the performance of our method, we consider the ensemble of generative classifiers not only from the penultimate features but also from other low-level features in DNNs. Formally, given training data, we extract $\ell$-th hidden features of DNNs, denoted by $f_\ell(\mathbf{x}) \in \mathbb{R}^{d_\ell}$, and compute the corresponding parameters of a generative classifier (i.e., $\widehat{\mu}_{\ell,c}$ and $\widehat{\boldsymbol{\Sigma}}_\ell$) using the (approximated version of) MCD estimator. Then, the final posterior distribution is obtained by the sum of all posterior distributions of generative classifiers:

$$\sum_\ell \alpha_\ell P\left(y = c | f_\ell(\mathbf{x})\right) = \sum_\ell \alpha_\ell \frac{\exp\left(\widehat{\mu}_{\ell,c}^\top \widehat{\boldsymbol{\Sigma}}_\ell^{-1} f_\ell(\mathbf{x}) - 0.5 \widehat{\mu}_{\ell,c}^\top \widehat{\boldsymbol{\Sigma}}_\ell^{-1} \widehat{\mu}_{\ell,c} + \log \widehat{\beta}_c\right)}{\sum_{c'} \exp\left(\widehat{\mu}_{\ell,c'}^\top \widehat{\boldsymbol{\Sigma}}_\ell^{-1} f_\ell(\mathbf{x}) - 0.5 \widehat{\mu}_{\ell,c'}^\top \widehat{\boldsymbol{\Sigma}}_\ell^{-1} \widehat{\mu}_{\ell,c'} + \log \widehat{\beta}_{c'}\right)},$$

where $\alpha_\ell$ is an ensemble weight at $\ell$-th layer. In our experiments, we choose the weight of each layer by optimizing negative log-likelihood (NLL) loss over the validation set. One can expect that this natural scheme can bring an extra gain in improving the performance due to ensemble effects.

To confirm that the proposed ensemble approach is indeed effective, we measure the classification accuracy of the generative classifier from different basic blocks of ResNet-34 (He et al., 2016) trained on CIFAR-10 dataset (Krizhevsky & Hinton, 2009) with various noise fractions, where the corresponding results on DenseNet models (Huang & Liu, 2017) can be found in Appendix C. For computational efficiency, the dimensions of the intermediate features are reduced using average pooling (see Section 3 for more details). First, we found that the performances of the generative classifiers from low-level features are more stable, while the accuracy of generative classifier from penultimate layer significantly decreases as the noisy fraction increases as shown in Figure 2(a). We expect that this is because the dimension (i.e., number of channels) of low-level features is usually smaller than that of high-level features. Since the breakdown point of MCD is inversely proportional to the feature dimension, the generative classifiers from low-level features can be more robust. This also coincides with the prior observation in the literature (Morcos et al., 2018) that DNNs tend to have similar hidden features at early layers, regardless of whether they train clean or noisy labels. More importantly, we found that generative classifiers from low-level features are more robust to strong adversarial attacks like CW (Carlini & Wagner, 2017), as shown in Figure 2(b). Therefore, we utilize the low-level generative classifiers as well to improve the generalization from noisy labels and robustness to adversarial attacks simultaneously.

## 3 EXPERIMENTAL RESULTS

In this section, we demonstrate the effectiveness of the proposed method using deep convolutional neural networks including DenseNet (Huang & Liu, 2017) and ResNet (He et al., 2016) on various vision datasets: CIFAR (Krizhevsky & Hinton, 2009) and SVHN (Netzer et al., 2011). Due to the space limit, we provide the more detailed experimental setups and results in Appendix B.

### 3.1 GENERALIZATION FROM NOISY LABELS

**Setup.** First, we evaluate the effectiveness of DDGC using deep models trained on datasets with noisy labels. We train DenseNet-100 and ResNet-34 for classifying CIFAR-10, CIFAR-100 and SVHN datasets. Similar to (Ma et al., 2018b; Han et al., 2018), we consider two types of noisy labels: corrupting a label to other class uniformly at random (uniform) and corrupting a label only to a specific class (flip). For ensembles of generative classifiers, we induce the generative classifiers from every end of basic block of DenseNet (or ResNet), where ensemble weights of each layer are tuned on a separate validation set, which consists of 500 images (i.e., only $\leq 1\%$ of the number of training samples) with clean labels.[1] Similar to (Lee et al., 2018), the size of feature maps on each convolutional layers is reduced by average pooling for computational efficiency: $\mathcal{F} \times \mathcal{H} \times \mathcal{W} \rightarrow \mathcal{F} \times 1$, where $\mathcal{F}$ is the number of channels and $\mathcal{H} \times \mathcal{W}$ is the spatial dimension.

**Verification of contributions from each technique.** We first evaluate the performance of generative classifiers with various assumptions: identity covariance (Euclidean) and tied covariance (LDA). In the case of identity covariance, we also apply a robust estimator called the least trimmed square (LTS) estimator (Rousseeuw, 1984) which finds a $K$-subset with smallest error and computes the sample mean from it, i.e., $\min_{\widehat{\mu}} \sum_{i=1}^{K} (\|\mathbf{x}_i - \widehat{\mu}\|_2^2)$. As shown in Figure 2(c), the generative classifiers with LDA assumption (blue and purple bars) generalize better than the generative classifiers with identity covariance (orange and green bars) well from noisy labels. Table 1 validates the contributions of the proposed techniques by incrementally applying our techniques to see the improvement from adding each component one by one. One can note that the generative classifier on features extracted from the penultimate layer outperforms the softmax classifier without the MCD estimator or ensemble method, while it still provides a comparable classification accuracy when the model is trained on clean dataset (i.e., noise = 0%). On top of that, by utilizing the MCD estimator, the classification accuracy is further improved compared to that employs only the naive sample estimator. This implies that the proposed method can improve the performance without any information

---

[1]For fair comparisons, one might also suggest fine-tuning the softmax classifier (with fixed features) using the validation data to improve the performance. However, the 1% data is often not enough for the purpose as the number of parameters of softmax classifier is too large. We indeed report the performance of the fine-tuned softmax classifier on Table 9 in Appendix D.

| Model | Inference method | Ensemble | Noise = 0% | 20% | 40% | 60% |
|---|---|---|---|---|---|---|
| DenseNet | Softmax | - | 94.42 | 80.24 | 68.61 | 53.34 |
| | Generative + sample | - | 94.31 | 85.08 | 74.72 | 59.49 |
| | | ✓ | 93.73 | 87.23 | 80.01 | 69.17 |
| | Generative + MCD (DDGC) | - | 94.37 | 85.96 | 78.34 | 66.16 |
| | | ✓ | 93.49 | **87.25** | **81.04** | **74.72** |
| ResNet | Softmax | - | 95.01 | 79.28 | 61.85 | 35.02 |
| | Generative + sample | - | 94.98 | 81.61 | 64.60 | 40.63 |
| | | ✓ | 94.98 | 87.23 | 78.40 | 61.94 |
| | Generative + MCD (DDGC) | - | 94.73 | 83.04 | 68.04 | 42.74 |
| | | ✓ | 94.32 | **87.25** | **80.01** | **71.06** |

Table 1: Effects of an ensemble method. We use the CIFAR-10 dataset with various uniform noise fractions. All values are percentages and the best results are highlighted in bold if the gain is bigger than 1%.

| Noise type (%) | ResNet | | | DenseNet | | |
|---|---|---|---|---|---|---|
| | CIFAR-10 | CIFAR-100 | SVHN | CIFAR-10 | CIFAR-100 | SVHN |
| | | Softmax / DDGC | | | Softmax / DDGC | |
| Clean | 95.01 / 94.32 | 77.51 / 76.55 | 95.96 / 96.09 | 94.42 / 93.49 | **76.41** / 73.65 | 96.59 / 96.18 |
| Uniform (20%) | 79.28 / **87.25** | 60.92 / **66.08** | 83.52 / **91.67** | 80.24 / **87.25** | 57.63 / **62.19** | 86.92 / **89.50** |
| Uniform (40%) | 61.85 / **80.01** | 44.08 / **59.72** | 72.89 / **87.16** | 68.61 / **81.04** | 45.08 / **53.98** | 81.91 / **85.71** |
| Uniform (60%) | 35.02 / **71.06** | 23.43 / **48.85** | 61.23 / **80.52** | 53.34 / **74.72** | 35.83 / **45.27** | 71.18 / **77.67** |
| Flip (20%) | 79.83 / **87.60** | 64.64 / **71.42** | 85.49 / **93.00** | 78.52 / **88.88** | 65.41 / **68.24** | 95.04 / 94.86 |
| Flip (40%) | 58.21 / **77.23** | 46.32 / **63.87** | 65.88 / **87.96** | 60.15 / **85.91** | 47.91 / **64.73** | 88.83 / **93.57** |

Table 2: Test accuracy (%) of different models trained on various datasets. We use the ensemble version of DDGC, and the best results are highlighted in bold if the gain is bigger than 1%.

of clean labels. In addition, the ensemble method significantly improves the classification accuracy. Finally, Table 2 reports the classification accuracy for all networks and datasets, where the proposed method significantly outperforms the softmax classifier for all tested cases.

**Compatibility and comparison with the state-of-art training methods.** We compare the performance of the standard softmax classifier and DDGC when they are combined with other various training methods for noisy environments, where more detailed explanations on them are given in Appendix B. We follow two experimental setups of Ma et al. (2018b)[2] and Han et al. (2018)[3]. The first setup considers the following methods training a single network: Hard/soft bootstrapping (Reed et al., 2014), forward/backward (Patrini et al., 2017), and D2L (Ma et al., 2018b). It uses ResNet-44 and only considers the uniform noise. The second setup considers the following methods training multiple networks, i.e., an ensemble of classifiers or a meta-learning model: Decoupling (Malach & Shalev-Shwartz, 2017), MentorNet (Jiang et al., 2018) and Co-teaching (Han et al., 2018). It uses a 9-layer CNN architecture, and considers the CIFAR-10 and CIFAR-100 datasets with uniform and flip noise types. Table 3 and 4 report the classification accuracy of softmax classifier and the ensemble version of DDGC, for the first and secon setups, respectively. They show that DDGC consistently outperforms the softmax inference under various training methods and noise types.[4]

## 3.2 ROBUSTNESS AGAINST ADVERSARIAL ATTACKS

**Setup.** We also evaluate if the proposed method can improve the robustness on adversarial attacks (Szegedy et al., 2014). It is well-known that the adversarial (visually imperceptible) perturbation to clean inputs can induce the DNNs to make incorrect predictions at test time. This undesirable property of DNNs has raised major security concerns. To verify that DDGC can improve the robustness to adversarial attacks, we train DenseNet-100 and ResNet-34 for classifying CIFAR-10, CIFAR-100 and SVHN datasets, and generate the adversarial samples using FGSM (Goodfellow et al., 2015), DeepFool (Moosavi Dezfooli et al., 2016) and CW (Carlini & Wagner, 2017) attacks, where the

---

[2]We used a reference implementation of Ma et al. (2018b): https://github.com/xingjunm/dimensionality-driven-learning.

[3]We used a reference implementation: https://github.com/bhanML/Co-teaching

[4]In the second setup, we only apply DDGC to a model pre-trained by Co-teaching because it outperforms other training methods.

| Dataset | Training method | Clean | Uniform (20%) | Uniform (40%) | Uniform (60%) |
|---|---|---|---|---|---|
| | | Softmax / DDGC | | | |
| CIFAR-10 | Cross-entropy | 94.34 / 94.29 | 81.95 / **86.80** | 63.84 / **78.75** | 62.45 / **69.06** |
| | Bootstrap (hard) | 94.56 / 94.49 | 82.90 / **87.26** | 75.97 / **81.81** | 72.91 / **75.61** |
| | Bootstrap (soft) | 94.46 / 94.31 | 80.29 / **85.20** | 65.22 / **77.86** | 58.55 / **69.71** |
| | Forward | 94.53 / 94.39 | 85.80 / **87.66** | 77.95 / **81.24** | 72.56 / **74.92** |
| | Backward | 94.39 / 94.45 | 77.44 / **81.38** | 62.83 / **72.76** | 56.64 / **65.95** |
| | D2L | 94.55 / 94.38 | 88.89 / **89.00** | 86.68 / 87.03 | 76.83 / **78.08** |
| CIFAR-100 | Cross-entropy | 76.31 / 75.87 | 61.11 / **66.33** | 45.08 / **58.68** | 34.97 / **44.96** |
| | Bootstrap (hard) | 75.65 / 75.49 | 61.61 / **65.14** | 51.27 / **57.61** | 39.04 / **47.24** |
| | Bootstrap (soft) | 76.40 / 75.88 | 60.28 / **65.54** | 47.66 / **57.97** | 34.68 / **44.84** |
| | Forward | 75.84 / 75.43 | 63.73 / **67.46** | 53.03 / **60.52** | 41.28 / **47.88** |
| | Backward | 76.75 / 76.28 | 56.24 / **62.13** | 37.70 / **51.82** | 23.55 / **39.16** |
| | D2L | 76.13 / 75.57 | 71.90 / 72.07 | 63.61 / **64.67** | 9.51 / **39.42** |
| SVHN | Cross-entropy | 96.38 / 96.15 | 83.45 / **91.46** | 60.86 / **83.40** | 38.29 / **68.71** |
| | Bootstrap (hard) | 96.40 / 96.26 | 83.43 / **92.24** | 74.25 / **88.04** | 66.51 / **82.03** |
| | Bootstrap (soft) | 96.51 / 96.09 | 86.43 / **91.25** | 58.22 / **83.90** | 44.52 / **73.11** |
| | Forward | 96.36 / 96.24 | 88.21 / **92.47** | 80.35 / **88.55** | 82.16 / **87.56** |
| | Backward | 96.43 / 96.31 | 87.00 / **87.31** | 72.02 / **78.65** | 50.54 / **71.06** |
| | D2L | 96.49 / 96.37 | 92.31 / **93.58** | 94.46 / 94.68 | 92.87 / 93.25 |

Table 3: Test accuracy (%) of ResNet trained on various training methods which utilize a single classifier. We use the ensemble version of DDGC, and the best results are highlighted in bold if the gain is bigger than 1%.

| Dataset | Noise type (%) | Cross-entropy | Decoupling | MentorNet | Co-teaching | Co-teaching + DDGC |
|---|---|---|---|---|---|---|
| CIFAR-10 | Flip (45%) | 49.50 | 48.80 | 58.14 | 71.17 | 70.50 |
| | Uniform (50%) | 48.87 | 51.49 | 71.10 | 74.12 | **76.26** |
| | Uniform (20%) | 76.25 | 80.44 | 80.76 | 82.13 | **84.49** |
| CIFAR-100 | Flip (45%) | 31.99 | 26.05 | 31.60 | 33.34 | **43.02** |
| | Uniform (50%) | 25.21 | 25.80 | 39.00 | 41.49 | **44.81** |
| | Uniform (20%) | 47.55 | 44.52 | 52.13 | 54.27 | **57.74** |

Table 4: Test accuracy (%) of 9-layer CNN trained on various training methods which utilize an ensemble of classifiers or meta-learning model. We use the ensemble version of DDGC and best results are highlighted in bold if the gain is bigger than 1%.

detailed explanations can be found in Appendix B. We consider the two types of adversarial attacks: generating the adversarial samples using a network, and then measuring their accuracy using the same (white-box) or another network (black-box). For all experiments, the adversarial samples are generated by targeting a softmax classifier or our generative classifier (see Appendix B for more details).

**Robustness against adversarial attacks.** Table 5 reports the classification accuracy of ResNet-34 on black-box adversarial attacks, and more results on white-box adversarial attacks and DenseNet-100 can be found in Appendix E. It shows that DDGC significantly improves the robustness against adversarial attacks when the training datasets contain noisy labels. We found that this is because the generative classifiers from low-level features are not utilized well, i.e., the trained weights (using validation) would be nearly zero for lower layers, when a training dataset only contains clean labels. Namely, the generative classifiers from low-level features are very robust (see Figure 2(b)), since the adversarial samples are generated in a way that mainly fools the upper layers of DNNs, i.e., thus both clean and adversarial samples produce similar hidden features at lower layers. We also apply DDGC to the "robust" learning models optimized by adversarial training methods (Goodfellow et al., 2015): generating FGSM samples and optimize the cross-entropy loss by treating them as additional training examples. Table 6 shows that DDGC further improves the robustness of deep models optimized by adversarial training.

| Dataset | Target | Adversarial attacks | Clean | Uniform (20%) | Uniform (40%) | Uniform (60%) |
|---------|--------|---------------------|-------|---------------|---------------|---------------|
| | | | Softmax / DDGC | | | |
| CIFAR-10 | Softmax | FGSM | 73.67 / 73.19 | 63.19 / **73.24** | 47.80 / **68.68** | 30.34 / **58.42** |
| | | DeepFool | 77.88 / 77.74 | 75.27 / **84.39** | 56.99 / **78.35** | 36.47 / **70.91** |
| | | CW | 36.36 / 36.86 | 65.45 / **76.26** | 50.68 / **73.59** | 33.08 / **67.81** |
| | DDGC | FGSM | 94.62 / 94.38 | 75.11 / **82.78** | 48.06 / **68.86** | 28.84 / **57.40** |
| | | DeepFool | 70.23 / 70.07 | 73.93 / **83.35** | 56.23 / **77.66** | 36.46 / **70.75** |
| | | CW | 13.00 / 12.50 | 28.40 / 28.40 | 17.20 / **19.40** | **12.70** / 10.70 |
| CIFAR-100 | Softmax | FGSM | **50.20** / 46.97 | 36.16 / **41.95** | 26.34 / **39.82** | 19.19 / **40.68** |
| | | DeepFool | **46.63** / 42.44 | 47.13 / **53.71** | 36.18 / **51.98** | 23.10 / **45.65** |
| | | CW | **35.66** / 33.50 | 38.57 / **45.63** | 32.21 / **47.61** | 22.52 / **44.74** |
| | DDGC | FGSM | **73.07** / 70.63 | 45.36 / **49.39** | 28.97 / **41.31** | 19.63 / **39.55** |
| | | DeepFool | **44.86** / 39.40 | 42.59 / **49.63** | 32.59 / **48.27** | 21.90 / **44.72** |
| | | CW | **26.81** / 24.13 | 15.54 / **17.04** | 10.59 / **17.95** | 8.81 / **22.90** |
| SVHN | Softmax | FGSM | 61.94 / 62.21 | 56.26 / **60.25** | 39.93 / **52.20** | 31.93 / **42.76** |
| | | DeepFool | 78.01 / 78.54 | 83.55 / **89.51** | 69.46 / **85.05** | 61.15 / **79.35** |
| | | CW | 57.27 / 58.11 | 76.21 / **83.85** | 60.64 / **79.56** | 51.73 / **73.83** |
| | DDGC | FGSM | 96.13 / 96.12 | 51.40 / **54.54** | 35.82 / **47.31** | 29.29 / **40.21** |
| | | DeepFool | 69.90 / 70.00 | 81.40 / **88.63** | 67.90 / **83.68** | 59.36 / **77.81** |
| | | CW | 36.10 / 36.80 | 46.30 / **51.20** | 34.30 / **36.70** | 30.00 / **33.80** |

Table 5: Test accuracy (%) of ResNet on black-box adversarial attacks. We use the ensemble version of DDGC, and the best results are highlighted in bold if the gain is bigger than 1%.

| Dataset | Target | Adversarial attacks | Clean | Uniform (20%) | Uniform (40%) | Uniform (60%) |
|---------|--------|---------------------|-------|---------------|---------------|---------------|
| | | | Softmax / DDGC | | | |
| CIFAR-10 | Softmax | FGSM | 91.63 / 91.59 | 79.01 / **84.54** | 62.47 / **78.45** | 37.71 / **66.96** |
| | | DeepFool | 93.72 / 93.34 | 80.95 / **85.83** | 63.79 / **79.29** | 38.23 / **68.04** |
| | | CW | 90.00 / 89.71 | 79.38 / **84.70** | 62.92 / **78.49** | 37.78 / **67.51** |
| | DDGC | FGSM | 94.81 / 94.55 | 80.82 / **86.08** | 62.22 / **78.62** | 37.22 / **66.81** |
| | | DeepFool | 93.06 / 92.65 | 80.66 / **85.64** | 63.72 / **79.23** | 38.32 / **68.01** |
| | | CW | 86.10 / 85.50 | 71.80 / **79.00** | 53.40 / **65.30** | 30.70 / **50.30** |
| CIFAR-100 | Softmax | FGSM | 63.67 / 62.74 | 51.39 / **58.88** | 35.78 / **52.85** | 22.13 / **44.86** |
| | | DeepFool | **60.92** / 59.29 | 53.34 / **60.94** | 37.10 / **54.90** | 23.47 / **46.35** |
| | | CW | 59.74 / 58.88 | 51.67 / **59.14** | 36.27 / **54.00** | 23.24 / **46.21** |
| | DDGC | FGSM | **70.17** / 69.08 | 54.25 / **61.23** | 36.81 / **52.98** | 21.97 / **44.22** |
| | | DeepFool | **59.81** / 57.90 | 52.72 / **61.27** | 36.90 / **54.86** | 23.81 / **46.45** |
| | | CW | **57.00** / 55.50 | 39.27 / **47.31** | 25.22 / **41.72** | 15.27 / **35.50** |
| SVHN | Softmax | FGSM | 68.00 / 68.34 | 68.08 / **72.33** | 51.56 / **62.60** | 39.01 / **54.11** |
| | | DeepFool | 85.68 / 85.77 | 87.98 / **91.63** | 74.69 / **86.48** | 65.64 / **81.32** |
| | | CW | 77.48 / 77.70 | 85.66 / **90.02** | 71.77 / **84.95** | 62.67 / **79.95** |
| | DDGC | FGSM | 96.36 / 96.27 | 64.40 / **67.44** | 47.48 / **58.00** | 35.90 / **51.89** |
| | | DeepFool | 79.54 / 79.36 | 85.59 / **89.90** | 71.95 / **84.81** | 63.36 / **79.86** |
| | | CW | 62.90 / 62.40 | 70.40 / **74.10** | 56.30 / **67.60** | 44.20 / **58.40** |

Table 6: Test accuracy (%) of ResNet optimized by adversarial training. We use the ensemble version of DDGC, and best results are highlighted in bold if the gain is bigger than 1%.

## 4 CONCLUSION

We propose a new inference method, easily applicable to any softmax neural classifier pre-trained on datasets with noisy labels. Our main idea is defining the generative classifier on top of fixed features from the pre-trained model. Such "deep generative classifiers" have been largely dismissed for fully-supervised classification settings as they are often substantially outperformed by discriminative deep classifiers (e.g., softmax classifiers). In contrast to this common belief, we show that it is possible to formulate a simple generative classifier that is more robust without sacrificing the discriminative performance. We expect that our work would bring a refreshing angle for other related tasks, e.g., adaptive attacks, memorization (Zhang et al., 2017) and interpretability (Morcos et al., 2018).

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

## A PRELIMINARIES

**Gaussian discriminant analysis**. In this section, we describe the basic concepts of the discriminative and generative classifier (Ng & Jordan, 2002). Formally, denote the random variable of the input and label as $\mathbf{x}$ and $y = \{1, \cdots, C\}$, respectively. For the classification task, the discriminative classifier directly defines a posterior distribution $P(y|\mathbf{x})$, i.e., learning a direct mapping between input $\mathbf{x}$ and label $y$. A popular model for discriminative classifier is softmax classifier which defines the posterior distribution as follows: $P\left(y = c|\mathbf{x}\right) = \frac{\exp\left(\mathbf{w}_c^\top \mathbf{x} + b_c\right)}{\sum_{c'} \exp\left(\mathbf{w}_{c'}^\top \mathbf{x} + b_{c'}\right)}$, where $\mathbf{w}_c$ and $b_c$ are weights and bias for a class $c$, respectively. In contrast to the discriminative classifier, the generative classifier defines the class conditional distribution $P(\mathbf{x}|y)$ and class prior $P(y)$ in order to indirectly define the posterior distribution by specifying the joint distribution $P(\mathbf{x}, y) = P(y) P(\mathbf{x}|y)$. Gaussian discriminant analysis (GDA) is a popular method to define the generative classifier by assuming that the class conditional distribution follows the multivariate Gaussian distribution and the class prior follows Bernoulli distribution: $P(\mathbf{x}|y = c) = \mathcal{N}(\mathbf{x}|\mu_c, \mathbf{\Sigma}_c), P(y = c) = \frac{\beta_c}{\sum_{c'} \beta_{c'}}$, where $\mu_c$ and $\mathbf{\Sigma}_c$ are the mean and covariance of multivariate Gaussian distribution, and $\beta_c$ is the unnormalized prior for class $c$. This classifier has been studied in various machine learning areas (e.g., semi-supervised learning (Lasserre et al., 2006) and incremental learning (Lee et al., 2018)).

In this paper, we focus on the special case of GDA, also known as the linear discriminant analysis (LDA). In addition to Gaussian assumption, LDA further assumes that all classes share the same covariance matrix, i.e., $\mathbf{\Sigma}_c = \mathbf{\Sigma}$. Since the quadratic term is canceled out with this assumption, the posterior distribution of generative classifier can be represented as follows:

$$P(y = c|\mathbf{x}) = \frac{P(y = c) P(\mathbf{x}|y = c)}{\sum_{c'} P(y = c') P(\mathbf{x}|y = c')} = \frac{\exp\left(\mu_c^\top \mathbf{\Sigma}^{-1} \mathbf{x} - \frac{1}{2}\mu_c^\top \mathbf{\Sigma}^{-1} \mu_c + \log \beta_c\right)}{\sum_{c'} \exp\left(\mu_{c'}^\top \mathbf{\Sigma}^{-1} \mathbf{x} - \frac{1}{2}\mu_{c'}^\top \mathbf{\Sigma}^{-1} \mu_{c'} + \log \beta_{c'}\right)}.$$

One can note that the above form of posterior distribution is equivalent to the softmax classifier by considering $\mu_c^\top \mathbf{\Sigma}^{-1}$ and $-\frac{1}{2}\mu_c^\top \mathbf{\Sigma}^{-1} \mu_c + \log \beta_c$ as its weight and bias, respectively. This implies that $\mathbf{x}$ might be fitted in Gaussian distribution during training a softmax classifier.

**Breakdown points**. The robustness of MCD estimator can be explained by the fact that it has high breakdown points (Hampel, 1971). Specifically, the breakdown point of an estimator measures the smallest fraction of observations that need to be replaced by arbitrary values to carry the estimate beyond all bounds. Formally, denote $\mathcal{Y}_M$ as a set obtained by replacing $M$ data points of set $\mathcal{Y}$ by some arbitrary values. Then, for a multivariate mean estimator $\mu = \mu(\mathcal{Y})$ from $\mathcal{Y}$, the breakdown point is defined as follows (see Appendix A for more detailed explanations including the breakdown point of covariance estimator):

$$\varepsilon^*(\mu, \mathcal{Y}) = \frac{1}{|\mathcal{Y}|} \min \left\{ M \in \{1, \cdots, |\mathcal{Y}|\} : \sup_{\mathcal{Y}_M} \|\mu(\mathcal{Y}) - \mu(\mathcal{Y}_M)\| = \infty \right\}.$$

For a multivariate estimator of covariance $\mathbf{\Sigma}$, we have

$$\varepsilon^*(\mathbf{\Sigma}, \mathcal{Y}) = \frac{1}{|\mathcal{Y}|} \min\{M \in \{1, \cdots, |\mathcal{Y}|\} : \sup_M \max_i \{|\log \lambda_i(\mathbf{\Sigma}(\mathcal{Y})) - \log \lambda_i(\mathbf{\Sigma}(\mathcal{Y}_M))|\}\},$$

where the $k-$th largest eigenvalue of a general $n \times n$ matrix is denoted by $\lambda_k(\mathbf{\Sigma})$, $k = 1, \cdots, n$ such that $\lambda_1(\mathbf{\Sigma}) \leq \lambda_2(\mathbf{\Sigma}) \leq \cdots \leq \lambda_n(\mathbf{\Sigma})$. This implies that we consider a covariance estimator to be broken whenever any of the eigenvalues can become arbitrary large or arbitrary close to 0.

## B EXPERIMENTAL SETUP

In this section, we describe detailed explanation about all the experiments described in Section 3.

**Detailed model architecture and datasets.** We consider two state-of-the-art neural network architectures: DenseNet (Huang & Liu, 2017) and ResNet (He et al., 2016). For DenseNet, our model follows the same setup as in Huang & Liu (2017): 100 layers, growth rate $k = 12$ and dropout rate 0. Also, we use ResNet with 34 and 44 layers, filters = 64 and dropout rate 0.[5] The softmax classifier

---

[5]ResNet architecture is available at `https://github.com/kuangliu/pytorch-cifar`.

is used, and each model is trained by minimizing the cross-entropy loss. We train DenseNet and ResNet for classifying CIFAR-10 (or 100) and SVHN datasets: the former consists of 50,000 training and 10,000 test images with 10 (or 100) image classes, and the latter consists of 73,257 training and 26,032 test images with 10 digits.[6] By following the experimental setup of Ma et al. (2018b), All networks were trained using SGD with momentum 0.9, weight decay $10^{-4}$ and an initial learning rate of 0.1. The learning rate is divided by 10 after epochs 40 and 80 for CIFAR-10/SVHN (120 epochs in total), and after epochs 80, 120 and 160 for CIFAR-100 (200 epochs in total). For our method, we extract the hidden features at $\{34, 46, 56, 67, 79, 89, 99\}$-th layers and $\{3, 7, 9, 11, 13, 15, 17, 19, 21, 23, 25, 27, 29, 31, 33\}$-th layers for DenseNet and ResNet, respectively.

**Training method for noisy label learning**. We consider the following training methods for noisy label learning:

(a) **Hard bootstrapping** (Reed et al., 2014): Training with new labels generated by a convex combination (the hard version) of the noisy labels and their predicted labels.

(b) **Soft bootstrapping** (Reed et al., 2014): Training with new labels generated by a convex combination (the soft version) of the noisy labels and their predictions.

(c) **Backward** (Patrini et al., 2017): Training via loss correction by multiplying the cross-entropy loss by a noise-aware correction matrix.

(d) **Forward** (Patrini et al., 2017): Training with label correction by multiplying the network prediction by a noise-aware correction matrix.

(e) **D2L** (Ma et al., 2018b): Training with new labels generated by a convex combination of the noisy labels and their predictions, where its weights are chosen by utilizing the Local Intrinsic Dimensionality (LID). (Ma et al., 2018a).

(f) **Decoupling** (Malach & Shalev-Shwartz, 2017): Updating the parameters only using the samples which have different prediction from two classifier.

(g) **MentorNet** (Jiang et al., 2018): An extra teacher network is pre-trained and then used to select clean samples for its student network.

(h) **Co-teaching** (Han et al., 2018): A simple ensemble method where each network selects its small-loss training data and back propagates the training data selected by its peer network.

(i) **Cross-entropy**: the conventional approach of training with cross-entropy loss.

**Adversarial attacks.** In this paper, we consider the following attack methods: fast gradient sign method (FGSM) (Goodfellow et al., 2015), DeepFool (Moosavi Dezfooli et al., 2016) and Carlini-Wagner (CW) (Carlini & Wagner, 2017). The FGSM directly perturbs normal input in the direction of the loss gradient. Formally, non-targeted adversarial samples are constructed as

$$\mathbf{x}_{adv} = \mathbf{x} + \varepsilon_{FGSM} \text{sign} \left( \bigtriangledown_{\mathbf{x}} \ell(y^*, P(y|\mathbf{x})) \right),$$

where $\varepsilon_{FGSM}$ is a magnitude of noise, $y^*$ is the prediction of classifier and $\ell$ is a loss function to measure the distance between the prediction and the ground truth. DeepFool works by finding the closest adversarial samples with geometric formulas. CW is an optimization-based method which arguably the most effective method. Formally, non-targeted adversarial samples are constructed as

$$\arg \min_{\mathbf{x}_{adv}} \lambda d(\mathbf{x}, \mathbf{x}_{adv}) - \ell(y^*, P(y|\mathbf{x}_{adv})),$$

where $\lambda$ is penalty parameter and $d(\cdot, \cdot)$ is a metric to quantify the distance between an original image and its adversarial counterpart. However, compared to FGSM, this method is much slower in practice. For all experiments, $L_2$ distance is used as a constraint. We used the library from (Guo et al., 2018) for generating adversarial samples.[7]

For all experiments, the adversarial samples are generated by targeting a softmax classifier or our generative classifier. Specifically, we generate the adversarial samples by attacking the ensemble of generative classifiers: $\sum_{\ell} \alpha_{\ell} P(y = c | f_{\ell}(\mathbf{x}))$ =

---

[6]We do not use the extra SVHN dataset for training.

[7]The code is available at https://github.com/facebookresearch/adversarial_image_defenses.

| Dataset | Target | Data type | Clean | | Uniform (20%) | | Uniform (40%) | | Uniform (60%) | |
|---|---|---|---|---|---|---|---|---|---|---|
| | | | $L_\infty$ | Acc. | $L_\infty$ | Acc. | $L_\infty$ | Acc. | $L_\infty$ | Acc. |
| CIFAR-10 | Softmax | Clean | 0 | 95.01 | 0 | 79.28 | 0 | 61.85 | 0 | 35.02 |
| | | FGSM | 0.05 | 56.90 | 0.05 | 25.26 | 0.05 | 23.71 | 0.05 | 20.88 |
| | | DeepFool | 0.34 | 0.36 | 0.06 | 0.04 | 0.03 | 0.00 | 0.02 | 0.00 |
| | | CW | 0.09 | 0.02 | 0.04 | 0.57 | 0.02 | 0.40 | 0.01 | 0.27 |
| | DDGC | Clean | 0 | 94.32 | 0 | 87.25 | 0 | 80.01 | 0 | 71.06 |
| | | FGSM | 0.05 | 94.59 | 0.05 | 78.40 | 0.05 | 59.33 | 0.05 | 52.87 |
| | | DeepFool | 0.44 | 5.25 | 0.09 | 66.26 | 0.04 | 71.77 | 0.02 | 68.76 |
| | | CW | 0.21 | 0.00 | 0.27 | 0.00 | 0.27 | 0.10 | 0.23 | 0.20 |
| CIFAR-100 | Softmax | Clean | 0 | 77.51 | 0 | 60.92 | 0 | 44.08 | 0 | 23.43 |
| | | FGSM | 0.05 | 31.78 | 0.05 | 17.95 | 0.05 | 13.20 | 0.05 | 10.11 |
| | | DeepFool | 0.33 | 0.27 | 0.12 | 0.08 | 0.06 | 0.05 | 0.04 | 0.01 |
| | | CW | 0.06 | 0.05 | 0.03 | 0.23 | 0.02 | 0.13 | 0.01 | 0.01 |
| | DDGC | Clean | 0 | 76.55 | 0 | 66.08 | 0 | 59.72 | 0 | 48.85 |
| | | FGSM | 0.05 | 72.44 | 0.05 | 42.25 | 0.05 | 34.06 | 0.05 | 31.31 |
| | | DeepFool | 0.32 | 8.50 | 0.14 | 28.40 | 0.09 | 30.90 | 0.07 | 35.86 |
| | | CW | 0.07 | 0.04 | 0.08 | 0.00 | 0.08 | 0.32 | 0.09 | 0.77 |
| SVHN | Softmax | Clean | 0 | 95.96 | 0 | 83.52 | 0 | 72.89 | 0 | 61.23 |
| | | FGSM | 0.20 | 54.96 | 0.20 | 42.58 | 0.20 | 24.61 | 0.20 | 24.42 |
| | | DeepFool | 0.52 | 1.30 | 0.09 | 0.14 | 0.06 | 0.13 | 0.05 | 0.17 |
| | | CW | 0.14 | 0.11 | 0.05 | 0.16 | 0.04 | 0.01 | 0.04 | 0.03 |
| | DDGC | Clean | 0 | 96.09 | 0 | 91.67 | 0 | 87.16 | 0 | 80.52 |
| | | FGSM | 0.20 | 96.15 | 0.20 | 47.26 | 0.20 | 41.23 | 0.20 | 35.46 |
| | | DeepFool | 0.77 | 17.16 | 0.10 | 71.75 | 0.07 | 80.08 | 0.05 | 75.00 |
| | | CW | 0.27 | 0.00 | 0.26 | 1.00 | 0.25 | 0.30 | 0.29 | 0.50 |

Table 7: The $L_\infty$ mean perturbation and classification accuracy of ResNet-34 on clean and adversarial samples.

| Dataset | Target | Data type | Clean | | Uniform (20%) | | Uniform (40%) | | Uniform (60%) | |
|---|---|---|---|---|---|---|---|---|---|---|
| | | | $L_\infty$ | Acc. | $L_\infty$ | Acc. | $L_\infty$ | Acc. | $L_\infty$ | Acc. |
| CIFAR-10 | Softmax | Clean | 0 | 94.42 | 0 | 80.24 | 0 | 68.61 | 0 | 53.34 |
| | | FGSM | 0.05 | 32.50 | 0.05 | 22.84 | 0.05 | 22.49 | 0.05 | 20.90 |
| | | DeepFool | 0.14 | 0.20 | 0.03 | 0.05 | 0.03 | 0.10 | 0.02 | 0.01 |
| | | CW | 0.06 | 0.08 | 0.03 | 0.39 | 0.03 | 0.27 | 0.02 | 0.18 |
| | DDGC | Clean | 0 | 93.49 | 0 | 87.25 | 0 | 81.04 | 0 | 74.72 |
| | | FGSM | 0.05 | 83.88 | 0.05 | 62.69 | 0.05 | 36.75 | 0.05 | 44.16 |
| | | DeepFool | 0.14 | 30.45 | 0.05 | 70.09 | 0.03 | 71.50 | 0.02 | 67.50 |
| | | CW | 0.12 | 0.00 | 0.11 | 0.04 | 0.12 | 0.09 | 0.23 | 0.54 |
| CIFAR-100 | Softmax | Clean | 0 | 76.41 | 0 | 57.63 | 0 | 45.08 | 0 | 35.83 |
| | | FGSM | 0.05 | 18.73 | 0.05 | 12.74 | 0.05 | 10.55 | 0.05 | 9.07 |
| | | DeepFool | 0.12 | 0.10 | 0.05 | 0.00 | 0.03 | 0.01 | 0.02 | 0.01 |
| | | CW | 0.03 | 0.22 | 0.02 | 0.20 | 0.02 | 0.20 | 0.01 | 0.12 |
| | DDGC | Clean | 0 | 73.65 | 0 | 62.19 | 0 | 53.98 | 0 | 45.27 |
| | | FGSM | 0.05 | 52.14 | 0.05 | 34.36 | 0.05 | 24.24 | 0.05 | 22.79 |
| | | DeepFool | 0.12 | 17.40 | 0.06 | 33.63 | 0.04 | 39.95 | 0.03 | 37.36 |
| | | CW | 0.07 | 0.18 | 0.08 | 1.04 | 0.07 | 2.31 | 0.06 | 3.86 |
| SVHN | Softmax | Clean | 0 | 96.59 | 0 | 86.92 | 0 | 81.91 | 0 | 71.18 |
| | | FGSM | 0.20 | 51.18 | 0.20 | 46.64 | 0.20 | 40.21 | 0.20 | 36.62 |
| | | DeepFool | 0.26 | 3.04 | 0.22 | 4.21 | 0.20 | 2.14 | 0.18 | 2.11 |
| | | CW | 0.12 | 0.15 | 0.12 | 0.31 | 0.10 | 0.16 | 0.10 | 0.09 |
| | DDGC | Clean | 0 | 96.18 | 0 | 89.50 | 0 | 85.71 | 0 | 77.67 |
| | | FGSM | 0.20 | 90.74 | 0.20 | 54.87 | 0.20 | 41.91 | 0.20 | 42.91 |
| | | DeepFool | 0.26 | 50.00 | 0.23 | 58.59 | 0.20 | 66.81 | 0.14 | 63.68 |
| | | CW | 0.25 | 0.00 | 0.33 | 1.59 | 0.39 | 4.63 | 0.46 | 11.22 |

Table 8: The $L_\infty$ mean perturbation and classification accuracy (%) of DenseNet-100 on clean and adversarial samples.

$$\sum_\ell \alpha_\ell \frac{\exp\left(\widehat{\mu}_{\ell,c}^\top \widehat{\Sigma}_\ell^{-1} f_\ell(\mathbf{x}) - 0.5\widehat{\mu}_{\ell,c}^\top \widehat{\Sigma}_\ell^{-1} \widehat{\mu}_{\ell,c} + \log \widehat{\beta}_c\right)}{\sum_{c'} \exp\left(\widehat{\mu}_{\ell,c'}^\top \widehat{\Sigma}_\ell^{-1} f_\ell(\mathbf{x}) - 0.5\widehat{\mu}_{\ell,c'}^\top \widehat{\Sigma}_\ell^{-1} \widehat{\mu}_{\ell,c'} + \log \widehat{\beta}_{c'}\right)}.$$ Here, we remark that a margin loss on the logit layer of each generative classifier is used in the case of CW attacks[8]. Due to time complexity, we only attack a generative classifier from a final layer in the case of DeepFool. Table 7 and 8 report the statistics of adversarial attacks including the $L_\infty$ mean perturbation and classification accuracy on adversarial attacks.

---

[8]Similar to Carlini & Wagner (2017), we use 2000 random samples.

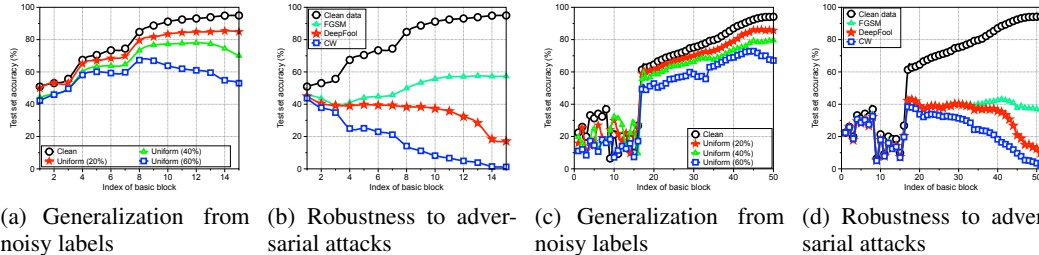

(a) Generalization from noisy labels  (b) Robustness to adversarial attacks  (c) Generalization from noisy labels  (d) Robustness to adversarial attacks

Figure 3: Layer-wise characteristics of generative classifiers from (a)/(b) ResNet-34 and (c)/(d) DenseNet-100 trained on the CIFAR-10 dataset.

| Model | Inference Method | Noisy Dataset | | | | | | | | |
| | | CIFAR-10 | | | CIFAR-100 | | | SVHN | | |
| | | 20% | 40% | 60% | 20% | 40% | 60% | 20% | 40% | 60% |
|---|---|---|---|---|---|---|---|---|---|---|
| ResNet | Softmax | 79.28 | 61.85 | 35.02 | 60.92 | 44.08 | 23.43 | 83.52 | 72.89 | 61.23 |
| | Softmax + fine-tuning (val) | 80.48 | 69.42 | 54.16 | 51.15 | 35.17 | 20.41 | 85.10 | 81.93 | 75.06 |
| | Generative + MCD | 83.04 | 68.04 | 42.74 | 63.20 | 51.71 | 33.96 | 89.60 | 78.88 | 66.96 |
| | Generative + MCD + ensemble (val) | **87.25** | **80.01** | **71.06** | **66.08** | **59.72** | **48.85** | **91.67** | **87.16** | **80.52** |
| DenseNet | Softmax | 80.24 | 68.61 | 53.34 | 57.63 | 45.08 | 35.83 | 86.92 | 81.91 | 71.18 |
| | Softmax + fine-tuning (val) | 82.47 | 74.38 | 63.25 | 48.69 | 35.97 | 27.73 | 86.83 | 76.39 | 68.13 |
| | Generative + MCD | 85.96 | 78.34 | 66.16 | 58.22 | 48.54 | 37.44 | 88.64 | 84.08 | 74.47 |
| | Generative + MCD + ensemble (val) | **87.25** | **81.04** | **74.72** | **62.19** | **53.98** | **45.27** | **89.50** | **85.71** | **77.67** |

Table 9: Comparison with softmax classifier fine-tuned with validation data. All values are percentages and the best results are highlighted in bold if gain is bigger than 1%.

## C    LAYER-WISE CHARACTERISTICS OF GENERATIVE CLASSIFIERS

Figure 3 shows the classification accuracy of the generative classifiers from different basic blocks of ResNet-34 (He et al., 2016) and DenseNet-101 (Huang & Liu, 2017). One can note that the generative classifiers from DenseNet and ResNet have different patterns due to the architecture design. In the case of DenseNet, we found that it produces meaning features after 20-th basic blocks. However, we remark that the performances of the generative classifiers from low-level features (from 20 to 40-th layers) of DenseNet are still more robust to adversarial attacks and noisy labels. Because of that, the ensemble of generative classifiers on DenseNet also improves the generalization from noisy labels and robustness to adversarial attacks.

## D    FINE-TUNING SOFTMAX LAYER USING VALIDATION SET

In this paper, we utilize a validation set, which consists of 500 images (i.e., only $\leq 1\%$ of the number of training samples) with clean labels. For fair comparisons, one might also suggest fine-tuning the softmax classifier (with fixed features) using the validation improve the classification accuracy. We indeed measure the classification accuracy of softmax classifier after fine-tuning in Table 9. One can note that the proposed method still outperforms softmax classifiers. Since 1% data is not enough compared to the number of softmax classifiers, fine-tuning can decrease the performance in the case of softmax classifier, while DDGC is still working well (it only trains the ensemble weights).

## E    MORE EXPERIMENTAL RESULTS ON ADVERSARIAL ATTACKS

In this section, we provide more experimental results on adversarial attacks. First, Table 10 and 11 show the classification accuracy on white-box adversarial attacks when we train ResNets using the

| Dataset | Attack type | Adversarial attacks | Clean | Uniform (20%) | Uniform (40%) | Uniform (60%) |
|---|---|---|---|---|---|---|
| | | | Softmax / DDGC | | | |
| CIFAR-10 | Softmax | FGSM | 56.90 / 57.51 | 25.26 / **57.11** | 23.71 / **59.32** | 20.88 / **54.97** |
| | | DeepFool | 0.36 / **18.48** | 0.04 / **71.14** | 0.00 / **73.21** | 0.00 / **68.92** |
| | | CW | 0.02 / **1.16** | 0.57 / **52.39** | 0.40 / **61.14** | 0.27 / **64.29** |
| | DDGC | FGSM | 94.74 / 94.59 | 70.68 / **78.40** | 25.54 / **59.33** | 14.95 / **52.87** |
| | | DeepFool | **11.13** / 5.25 | 4.71 / **66.26** | 3.20 / **71.77** | 4.17 / **68.76** |
| | | CW | 0.00 / 0.00 | 0.20 / 0.00 | 0.20 / 0.10 | 0.20 / 0.20 |
| CIFAR-100 | Softmax | FGSM | 31.78 / 31.94 | 17.95 / **30.23** | 13.20 / **30.81** | 10.11 / **35.62** |
| | | DeepFool | 0.27 / **9.84** | 0.08 / **37.07** | 0.05 / **42.15** | 0.01 / **40.82** |
| | | CW | 0.05 / **5.07** | 0.23 / **27.95** | 0.13 / **35.32** | 0.01 / **39.90** |
| | DDGC | FGSM | **74.51** / 72.44 | 35.41 / **42.25** | 23.68 / **34.06** | 12.08 / **31.31** |
| | | DeepFool | 4.09 / **8.50** | 4.72 / **28.40** | 3.54 / **30.90** | 1.63 / **35.86** |
| | | CW | 0.95 / 0.04 | 0.86 / 0.00 | 0.31 / 0.32 | 0.22 / 0.77 |
| SVHN | Softmax | FGSM | 54.96 / **56.46** | 42.58 / **54.26** | 24.61 / **46.80** | 24.42 / **40.92** |
| | | DeepFool | 1.30 / **45.40** | 0.14 / **79.29** | 0.13 / **81.59** | 0.17 / **75.87** |
| | | CW | 0.11 / **13.39** | 0.16 / **56.58** | 0.01 / **68.22** | 0.03 / **61.93** |
| | DDGC | FGSM | 96.19 / 96.15 | 40.82 / **47.26** | 19.80 / **41.23** | 18.13 / **35.46** |
| | | DeepFool | 13.00 / **17.16** | 17.58 / **71.75** | 19.66 / **80.08** | 27.83 / **75.00** |
| | | CW | 0.10 / 0.00 | 0.70 / 1.00 | 0.40 / 0.30 | 0.10 / 0.50 |

Table 10: Test accuracy (%) of ResNet on white-box adversarial attacks. We use the ensemble version of DDGC, and the best results are highlighted in bold if gain is bigger than 1%.

| Dataset | Target | Adversarial attacks | Clean | Uniform (20%) | Uniform (40%) | Uniform (60%) |
|---|---|---|---|---|---|---|
| | | | Softmax / DDGC | | | |
| CIFAR-10 | Softmax | FGSM | 74.72 / 74.53 | 35.21 / **64.06** | 24.64 / **67.93** | 20.70 / **59.87** |
| | | DeepFool | 1.09 / **14.45** | 0.04 / **61.54** | 0.05 / **69.22** | 0.00 / **61.40** |
| | | CW | 0.05 / **1.31** | 0.00 / **40.81** | 0.00 / **59.86** | 0.00 / **57.22** |
| | DDGC | FGSM | 94.68 / 94.44 | 59.01 / **73.32** | 31.60 / **67.96** | 12.83 / **55.54** |
| | | DeepFool | 9.41 / **10.77** | 9.77 / **44.86** | 9.68 / **66.50** | 9.36 / **60.40** |
| | | CW | 0.14 / 0.04 | 0.04 / 0.00 | 0.18 / 0.04 | 0.31 / 0.00 |
| CIFAR-100 | Softmax | FGSM | 37.73 / **42.57** | 24.04 / **47.00** | 16.52 / **46.46** | 11.52 / **41.62** |
| | | DeepFool | 0.13 / **10.40** | 0.04 / **36.09** | 0.00 / **42.95** | 0.00 / **41.45** |
| | | CW | 0.00 / **9.31** | 0.04 / **34.81** | 0.00 / **44.04** | 0.00 / **41.54** |
| | DDGC | FGSM | 62.60 / 62.88 | 40.98 / **51.68** | 25.35 / **43.29** | 10.92 / **36.39** |
| | | DeepFool | 2.81 / **6.22** | 3.45 / **21.36** | 2.31 / **27.36** | 1.86 / **34.13** |
| | | CW | **1.18** / 0.00 | 0.81 / 0.00 | 0.45 / 0.04 | 0.27 / 0.04 |
| SVHN | Softmax | FGSM | 57.86 / **60.31** | 44.25 / **66.10** | 30.17 / **60.10** | 27.59 / **56.45** |
| | | DeepFool | 2.54 / **35.72** | 0.18 / **77.50** | 0.13 / **79.63** | 0.00 / **75.40** |
| | | CW | 0.50 / **19.40** | 0.04 / **63.68** | 0.04 / **69.13** | 0.00 / **66.90** |
| | DDGC | FGSM | 96.42 / 96.35 | 50.07 / **61.63** | 22.10 / **52.05** | 18.79 / **51.15** |
| | | DeepFool | 15.81 / **21.81** | 20.95 / **73.31** | 18.18 / **78.09** | 33.36 / **76.27** |
| | | CW | 0.27 / 0.63 | 0.77 / 0.90 | 0.36 / 0.54 | 1.18 / 1.40 |

Table 11: Test accuracy (%) of ResNet optimized by adversarial training on CIFAR-10. We test the white-box adversarial attacks. We use the ensemble version of DDGC, and best results are highlighted in bold if the gain is bigger than 1%.

cross-entropy loss and adversarial training, respectively. We also report the classification accuracy of DenseNet on white-box and black-box adversarial attacks in Table 12 and 13, respectively.

| Dataset | Target | Adversarial attacks | Clean | Uniform (20%) | Uniform (40%) | Uniform (60%) |
|---|---|---|---|---|---|---|
| | | | Softmax / DDGC | | | |
| CIFAR-10 | Softmax | FGSM | 32.50 / **38.00** | 22.84 / **56.05** | 22.49 / **46.51** | 20.90 / **52.63** |
| | | DeepFool | 0.20 / **36.09** | 0.05 / **76.94** | 0.10 / **75.03** | 0.01 / **68.83** |
| | | CW | 0.08 / **4.27** | 0.39 / **55.15** | 0.27 / **48.83** | 0.18 / **54.83** |
| | DDGC | FGSM | 83.92 / 83.88 | 53.33 / **62.69** | 20.51 / **36.75** | 13.97 / **44.16** |
| | | DeepFool | 30.81 / 30.45 | 6.95 / **70.09** | 12.22 / **71.50** | 13.68 / **67.50** |
| | | CW | 0.00 / 0.00 | 0.13 / 0.04 | 0.22 / 0.09 | 0.63 / 0.54 |
| CIFAR-100 | Softmax | FGSM | 18.73 / **22.65** | 12.74 / **27.18** | 10.55 / **22.36** | 9.07 / **22.09** |
| | | DeepFool | 0.10 / **19.31** | 0.00 / **40.09** | 0.01 / **42.10** | 0.01 / **37.46** |
| | | CW | 0.22 / **12.87** | 0.20 / **30.14** | 0.20 / **32.20** | 0.12 / **31.46** |
| | DDGC | FGSM | **54.25** / 52.14 | 30.14 / **34.26** | 19.67 / **24.24** | 12.46 / **22.79** |
| | | DeepFool | 15.27 / **17.40** | 8.77 / **33.63** | 6.45 / **39.95** | 7.90 / **37.36** |
| | | CW | **2.45** / 0.18 | 1.09 / 1.04 | 0.90 / **2.31** | 1.00 / **3.86** |
| SVHN | Softmax | FGSM | 51.18 / **53.75** | 46.64 / **50.98** | 40.21 / **46.60** | 36.62 / **40.05** |
| | | DeepFool | 3.04 / **66.30** | 4.21 / **71.28** | 2.14 / **71.97** | 2.11 / **62.26** |
| | | CW | 0.15 / **27.01** | 0.31 / **36.48** | 0.16 / **33.89** | 0.09 / **26.86** |
| | DDGC | FGSM | 90.39 / 90.74 | 52.14 / **54.87** | 36.31 / **41.91** | 41.18 / **42.91** |
| | | DeepFool | 41.63 / **50.00** | 26.40 / **58.59** | 36.27 / **66.81** | 52.54 / **63.68** |
| | | CW | **1.45** / 0.00 | 1.40 / 1.59 | 3.31 / **4.63** | 11.36 / 11.22 |

Table 12: Test accuracy (%) of DenseNet on white-box adversarial attacks. We use the ensemble version of DDGC, and the best results are highlighted in bold if gain is bigger than 1%.

| Dataset | Target | Adversarial attacks | Clean | Uniform (20%) | Uniform (40%) | Uniform (60%) |
|---|---|---|---|---|---|---|
| | | | Softmax / DDGC | | | |
| CIFAR-10 | Softmax | FGSM | 62.96 / **64.66** | 60.63 / **70.19** | 47.77 / **55.11** | 40.07 / **62.02** |
| | | DeepFool | 87.68 / 87.78 | 76.48 / **84.54** | 67.93 / **80.38** | 51.31 / **73.57** |
| | | CW | 52.12 / **54.92** | 63.48 / **74.23** | 55.41 / **67.45** | 45.03 / **69.03** |
| | DDGC | FGSM | 87.69 / 87.82 | 63.25 / **70.94** | 44.47 / **51.43** | 38.14 / **59.76** |
| | | DeepFool | 88.31 / 88.36 | 75.81 / **84.31** | 66.04 / **79.86** | 52.36 / **73.68** |
| | | CW | 24.31 / 24.22 | 26.77 / **31.04** | 19.68 / 19.45 | 15.36 / 15.81 |
| CIFAR-100 | Softmax | FGSM | **48.84** / 45.83 | 35.07 / **39.32** | 24.60 / **27.43** | 21.64 / **28.72** |
| | | DeepFool | **66.87** / 62.53 | 53.00 / **56.05** | 43.09 / **48.18** | 32.42/ **41.71** |
| | | CW | **55.74** / 53.09 | 44.70 / **49.49** | 36.49 / **41.53** | 29.67 / **38.97** |
| | DDGC | FGSM | **61.21** / 57.29 | 39.52 / **42.20** | 26.35 / **28.31** | 22.82 / **29.00** |
| | | DeepFool | **68.04** / 64.95 | 52.04 / **55.86** | 43.27 / **48.36** | 33.59 / **42.31** |
| | | CW | **37.86** / 34.09 | 23.31 / **26.81** | 18.63 / **20.81** | 17.54 / **24.68** |
| SVHN | Softmax | FGSM | 58.08 / **59.10** | 53.76 / **56.76** | 47.58 / **52.08** | 46.82 / **47.91** |
| | | DeepFool | 89.59 / 90.13 | 86.60 / **88.25** | 82.60 / **85.95** | 81.63 / **82.81** |
| | | CW | 66.35 / **68.61** | 68.27 / **71.23** | 60.25 / **66.23** | 64.19 / **66.20** |
| | DDGC | FGSM | 89.87 / 90.23 | 52.57 / **55.92** | 43.59 / **47.60** | 46.66 / **47.91** |
| | | DeepFool | 87.40 / **88.04** | 81.18 / **83.95** | 79.90 / **83.72** | 78.81 / **80.31** |
| | | CW | 40.59 / 41.50 | 39.68 / **41.18** | 38.59 / **39.77** | 45.45 / **46.63** |

Table 13: Test accuracy (%) of DenseNet on black-box adversarial attacks. We use the ensemble version of DDGC, and the best results are highlighted in bold if gain is bigger than 1%.

# F  PROOF OF THEOREM 1

In this section, we present a proof of Theorem 1, which consists of three statements: the limit of estimated error ratio (3), estimation error ratio (4) and mahalanobis distance (5). We prove each statement one by one as stated in below. For convenience, we skip to mention the Continuous Mapping Theorem[9] and the number of training samples $N$ goes to infinity for all convergences in the proof.

## F.1  PROOF OF THE LIMIT OF ESTIMATION ERROR (4)

We start with the following lemma, which shows the convergences of sample and MCD estimators with single Gaussian distribution as the number of training samples $N$ goes to infinity.

---

[9]P. Billingsley, Convergence of Probability Measures, John Wiley & Sons, 1999

**Lemma 1.** *Suppose we have $N$ number of $d$-dimensional training samples $\mathcal{X}_N = \{\mathbf{x}_1, \cdots, \mathbf{x}_N\}$ and $\mathcal{X}_N$ contains outlier samples with the fixed fraction $\delta_{\text{out}} < 1$. We assume the outlier samples are from an arbitrary distribution $P_{\text{out}}$ with zero mean and finite covariance matrix $\sigma_{\text{out}}^2 \mathbf{I}$, and the other samples are from a multivariate Gaussian distribution $P_{\text{data}}$ with mean $\mu$ and covariance matrix $\sigma^2 \mathbf{I}$. Let $\bar{\mu}$ and $\bar{\Sigma}$ be the mean and covariance matrix of sample estimator, and let $\widehat{\mu}$ and $\widehat{\Sigma}$ be the mean and covariance matrix of MCD estimator which selects samples from $\mathcal{X}_N$ with the fixed fraction $\frac{d}{N} < \delta_{\text{mcd}} < 1$ to optimize its objective (2). Then the mean and covariance matrix of sample estimator converge almost surely to below as $N \to \infty$:*

$$\bar{\mu} \overset{\text{a.s.}}{\to} (1 - \delta_{\text{out}}) \mu, \quad \bar{\Sigma} \overset{\text{a.s.}}{\to} \left((1 - \delta_{\text{out}}) \sigma^2 + \delta_{\text{out}} \sigma_{\text{out}}^2\right) \mathbf{I} + \delta_{\text{out}} (1 - \delta_{\text{out}}) \mu \mu^T.$$

*In addition, if $\delta_{\text{mcd}} \leq 1 - \delta_{\text{out}}$ and $\sigma^2 < \sigma_{\text{out}}^2$, the mean and covariance matrix of MCD estimator converge almost surely to below as $N \to \infty$:*

$$\widehat{\mu} \overset{\text{a.s.}}{\to} \mu, \qquad \widehat{\Sigma} \overset{\text{a.s.}}{\to} \sigma^2 \mathbf{I}.$$

A proof of the above lemma is given in appendix F.4, where it is built upon the fact that the determinant of covariance matrix with some assumptions can be expressed as the $d$-th degree polynomial of outlier ratio.

Lemma 1 states the convergences of sample and MCD estimators on single multivariate Gaussian distribution. One can extend it to $C$ number of multivariate Gaussian distributions, which have the class mean $\mu_c$ and class covariance matrix $\Sigma_c$ on each class label $c \in \{1, ..., C\}$, by the assumptions $\mathcal{A}1 \sim \mathcal{A}5$. Then one can induce the class mean of MCD and sample estimators converge almost surely as follows:

$$\widehat{\mu}_c \overset{\text{a.s.}}{\to} \mu_c, \qquad \bar{\mu}_c \overset{\text{a.s.}}{\to} (1 - \delta_{\text{out}}) \mu_c,$$

which implies that

$$\|\mu_c - \widehat{\mu}_c\|_1 \overset{\text{a.s.}}{\to} 0, \qquad \|\mu_c - \bar{\mu}_c\|_1 \overset{\text{a.s.}}{\to} \delta_{\text{out}} \|\mu_c\|_1.$$

This completes the proof of the limit of estimation error (4).

## F.2    PROOF OF THE LIMIT OF MAHALANOBIS DISTANCE RATIO (5)

From the assumptions $\mathcal{A}1$, all class covariance matrices are the same, i.e., $\Sigma_c = \sigma^2 \mathbf{I}$. Then tied covariance matrices $\bar{\Sigma}$ and $\widehat{\Sigma}$ are given by gathering $\bar{\Sigma}_c$ and $\widehat{\Sigma}_c$ on each class $c$ respectively:

$$\bar{\Sigma} = \frac{\sum_c N_c \bar{\Sigma}_c}{\sum_c N_c} = \frac{\sum_c \bar{\Sigma}_c}{C}, \quad \widehat{\Sigma} = \frac{\sum_c K_c \widehat{\Sigma}_c}{\sum_c K_c} = \frac{\sum_c \widehat{\Sigma}_c}{C}. \tag{6}$$

From the tied covariance matrices (6) and Lemma 1, one can induce their convergences and limits as follow:

$$\bar{\Sigma} \overset{\text{a.s.}}{\to} \left((1 - \delta_{\text{out}}) \sigma^2 + \delta_{\text{out}} \sigma_{\text{out}}^2\right) \mathbf{I} + \delta_{\text{out}} (1 - \delta_{\text{out}}) \frac{1}{C} \sum_c \mu_c \mu_c^T, \tag{7}$$

$$\widehat{\Sigma} \overset{\text{a.s.}}{\to} \sigma^2 \mathbf{I}.$$

Since the assumption $\mathcal{A}2$ gives a diagonal matrix $\mathbf{D} = \frac{1}{C} \sum_c \mu_c \mu_c^T$, the limit of covariance matrix of sample estimator (7) and its inverse are also diagonal matrices. Then the limit of inverse matrix $\bar{\Sigma}^{-1}$ is given as follows:

$$\bar{\Sigma}^{-1} \overset{\text{a.s.}}{\to} \left(\left((1 - \delta_{\text{out}}) \sigma^2 + \delta_{\text{out}} \sigma_{\text{out}}^2\right) \mathbf{I} + \delta_{\text{out}} (1 - \delta_{\text{out}}) \mathbf{D}\right)^{-1}. \tag{8}$$

Since the limit of inverse matrix $\bar{\Sigma}^{-1}$ (8) is a diagonal matrix, the limit of the mahalanobis distance between $\bar{\mu}_c$ and $\bar{\mu}_{c'}$ is given as follows:

$$(\bar{\mu}_c - \bar{\mu}_{c'})^T \bar{\Sigma}^{-1} (\bar{\mu}_c - \bar{\mu}_{c'}) = \mathbf{tr}\left(\bar{\Sigma}^{-1} (\bar{\mu}_c - \bar{\mu}_{c'})(\bar{\mu}_c - \bar{\mu}_{c'})^T\right)$$

$$\overset{\text{a.s.}}{\to} \sum_i \frac{(1 - \delta_{\text{out}})^2 (\mu_{ci} - \mu_{c'i})^2}{(1 - \delta_{\text{out}}) \sigma^2 + \delta_{\text{out}} \sigma_{\text{out}}^2 + \delta_{\text{out}} (1 - \delta_{\text{out}}) D_i}, \tag{9}$$

where $\mathbf{D} = \text{diag}(D_1, ..., D_d)$. By the assumption $\mathcal{A}4$, the denominator term of (9) is greater than $\sigma^2$ for all $i$, i.e.,

$$(1 - \delta_{\text{out}})\sigma^2 + \delta_{\text{out}}\sigma_{\text{out}}^2 + \delta_{\text{out}}(1 - \delta_{\text{out}})D_i \geq \sigma^2 \text{ for } \forall i.$$

Then the limit of mahalanobis distance between mean of sample estimators (9) has the following upper bound:

$$(\bar{\mu}_c - \bar{\mu}_{c'})^T \bar{\boldsymbol{\Sigma}}^{-1}(\bar{\mu}_c - \bar{\mu}_{c'}) \xrightarrow{\text{a.s.}} \sum_i \frac{(1 - \delta_{\text{out}})^2 (\mu_{ci} - \mu_{c'i})^2}{(1 - \delta_{\text{out}})\sigma^2 + \delta_{\text{out}}\sigma_{\text{out}}^2 + \delta_{\text{out}}(1 - \delta_{\text{out}})D_i}$$

$$\leq (1 - \delta_{\text{out}})^2 \frac{(\mu_i - \mu_c)^T(\mu_i - \mu_c)}{\sigma^2}. \tag{10}$$

One can also induce the limit of the mahalanobis distance between mean of MCD estimators $\widehat{\mu}_c$ and $\widehat{\mu}_{c'}$ as follows:

$$(\widehat{\mu}_i - \widehat{\mu}_c)^T \widehat{\boldsymbol{\Sigma}}^{-1}(\widehat{\mu}_i - \widehat{\mu}_c) = \mathbf{tr}\left(\widehat{\boldsymbol{\Sigma}}^{-1}(\widehat{\mu}_i - \widehat{\mu}_c)(\widehat{\mu}_i - \widehat{\mu}_c)^T\right) \xrightarrow{\text{a.s.}} \frac{(\mu_i - \mu_c)^T(\mu_i - \mu_c)}{\sigma^2}. \tag{11}$$

From the limits of mahalanobis distance (10) and (11), the limit of mahalnobis distance ratio (5) is induced as follows:

$$\frac{(\bar{\mu}_c - \bar{\mu}_{c'})^T \bar{\boldsymbol{\Sigma}}^{-1}(\bar{\mu}_c - \bar{\mu}_{c'})}{(\widehat{\mu}_c - \widehat{\mu}_{c'})^T \widehat{\boldsymbol{\Sigma}}^{-1}(\widehat{\mu}_c - \widehat{\mu}_{c'})} \xrightarrow{\text{a.s.}} \frac{\sum_i \frac{(1 - \delta_{\text{out}})^2 (\mu_{ci} - \mu_{c'i})^2}{(1 - \delta_{\text{out}})\sigma^2 + \delta_{\text{out}}\sigma_{\text{out}}^2 + \delta_{\text{out}}(1 - \delta_{\text{out}})D_i}}{\frac{(\mu_i - \mu_c)^T(\mu_i - \mu_c)}{\sigma^2}}$$

$$\leq \frac{(1 - \delta_{\text{out}})^2 \frac{(\mu_i - \mu_c)^T(\mu_i - \bar{\mu}_c)}{\sigma^2}}{\frac{(\mu_i - \mu_c)^T(\mu_i - \mu_c)}{\sigma^2}} = (1 - \delta_{\text{out}})^2 \leq 1.$$

This completes the proof of the limit of mahalnobis distance ratio (5).

### F.3 PROOF OF THE LIMIT OF ESTIMATED ERROR RATIO (3)

Remind that the estimated error $B(\widehat{\mu}_c, \widehat{\mu}_{c'}, \boldsymbol{\Sigma}, \widehat{\boldsymbol{\Sigma}})$ is defined as follows:

$$B(\widehat{\mu}_c, \widehat{\mu}_{c'}, \boldsymbol{\Sigma}, \widehat{\boldsymbol{\Sigma}}) := \exp\left(-\frac{1}{8} \frac{\left[(\widehat{\mu}_c - \widehat{\mu}_{c'})^T \widehat{\boldsymbol{\Sigma}}^{-1}(\widehat{\mu}_c - \widehat{\mu}_{c'})\right]^2}{(\widehat{\mu}_{c'} - \widehat{\mu}_c)^T \widehat{\boldsymbol{\Sigma}}^{-1} \boldsymbol{\Sigma} \widehat{\boldsymbol{\Sigma}}^{-1}(\widehat{\mu}_{c'} - \widehat{\mu}_c)}\right). \tag{12}$$

By the property of trace operator, the inner term of estimated error (12) is equal to

$$\frac{[(\widehat{\mu}_c - \widehat{\mu}_{c'})^T \widehat{\boldsymbol{\Sigma}}^{-1}(\widehat{\mu}_c - \widehat{\mu}_{c'})]^2}{(\widehat{\mu}_{c'} - \widehat{\mu}_c)^T \widehat{\boldsymbol{\Sigma}}^{-1} \boldsymbol{\Sigma} \widehat{\boldsymbol{\Sigma}}^{-1}(\widehat{\mu}_{c'} - \widehat{\mu}_c)} = \frac{\mathbf{tr}\left(\widehat{\boldsymbol{\Sigma}}^{-1}(\widehat{\mu}_c - \widehat{\mu}_{c'})(\widehat{\mu}_c - \widehat{\mu}_{c'})^T\right)^2}{\mathbf{tr}\left(\widehat{\boldsymbol{\Sigma}}^{-1} \boldsymbol{\Sigma} \widehat{\boldsymbol{\Sigma}}^{-1}(\widehat{\mu}_c - \widehat{\mu}_{c'})(\widehat{\mu}_c - \widehat{\mu}_{c'})^T\right)}. \tag{13}$$

From the right hand side of (13), the negative logarithms of estimated error (12) of sample and MCD estimators converge as follows:

$$-8\log(B(\bar{\mu}_c, \bar{\mu}_{c'}, \boldsymbol{\Sigma}, \bar{\boldsymbol{\Sigma}})) = \frac{\mathbf{tr}\left(\bar{\boldsymbol{\Sigma}}^{-1}(\bar{\mu}_c - \bar{\mu}_{c'})(\bar{\mu}_c - \bar{\mu}_{c'})^T\right)^2}{\mathbf{tr}\left(\bar{\boldsymbol{\Sigma}}^{-1}\sigma^2\mathbf{I}\bar{\boldsymbol{\Sigma}}^{-1}(\bar{\mu}_c - \bar{\mu}_{c'})(\bar{\mu}_c - \bar{\mu}_{c'})^T\right)}$$

$$\xrightarrow{\text{a.s.}} \frac{(1 - \delta_{\text{out}})^2}{\sigma^2} \frac{\left(\sum_i \frac{(\mu_{ci} - \mu_{c'i})^2}{(1 - \delta_{\text{out}})\sigma^2 + \delta_{\text{out}}\sigma_{\text{out}}^2 + \delta_{\text{out}}(1 - \delta_{\text{out}})D_i}\right)^2}{\sum_i \frac{(\mu_{ci} - \mu_{c'i})^2}{((1 - \delta_{\text{out}})\sigma^2 + \delta_{\text{out}}\sigma_{\text{out}}^2 + \delta_{\text{out}}(1 - \delta_{\text{out}})D_i)^2}}, \tag{14}$$

$$-8\log(B(\widehat{\mu}_c, \widehat{\mu}_{c'}, \boldsymbol{\Sigma}, \widehat{\boldsymbol{\Sigma}})) \xrightarrow{\text{a.s.}} \frac{\mathbf{tr}\left((\sigma^2\mathbf{I})^{-1}(\mu_c - \mu_{c'})(\mu_c - \mu_{c'})^T\right)^2}{\mathbf{tr}\left((\sigma^2\mathbf{I})^{-1}\sigma^2\mathbf{I}(\sigma^2\mathbf{I})^{-1}(\mu_c - \mu_{c'})(\mu_c - \widehat{\mu}_{c'})^T\right)}$$

$$= \frac{(\mu_c - \mu_{c'})^T(\mu_c - \mu_{c'})}{\sigma^2}. \tag{15}$$

The relation between the limits of (14) and (15) can be induced by the Cauchy-Schwarz inequality,

$$\sum_i (\mu_{ci} - \mu_{c'i})^2 \sum_i \left( \frac{\mu_{ci} - \mu_{c'i}}{(1-\delta_{\text{out}})\sigma^2 + \delta_{\text{out}}\sigma_{\text{out}}^2 + \delta_{\text{out}}(1-\delta_{\text{out}})D_i} \right)^2$$

$$\geq \left( \sum_i \frac{(\mu_{ci} - \mu_{c'i})^2}{(1-\delta_{\text{out}})\sigma^2 + \delta_{\text{out}}\sigma_{\text{out}}^2 + \delta_{\text{out}}(1-\delta_{\text{out}})D_i} \right)^2.$$

The above inequality implies

$$(\mu_c - \mu_{c'})^T (\mu_c - \mu_{c'}) \geq \frac{\left( \sum_i \frac{(\mu_{ci}-\mu_{c'i})^2}{(1-\delta_{\text{out}})\sigma^2 + \delta_{\text{out}}\sigma_{\text{out}}^2 + \delta_{\text{out}}(1-\delta_{\text{out}})D_i} \right)^2}{\sum_i \frac{(\mu_{ci}-\mu_{c'i})^2}{((1-\delta_{\text{out}})\sigma^2 + \delta_{\text{out}}\sigma_{\text{out}}^2 + \delta_{\text{out}}(1-\delta_{\text{out}})D_i)^2}}.$$

From the fact that $(1-\delta_{\text{out}})^2 \leq 1$, the limits of (14) and (15) hold the following inequality:

$$\frac{(\mu_c - \mu_{c'})^T (\mu_c - \mu_{c'})}{\sigma^2} \geq \frac{(1-\delta_{\text{out}})^2}{\sigma^2} \frac{\left( \sum_i \frac{(\mu_{ci}-\mu_{c'i})^2}{(1-\delta_{\text{out}})\sigma^2 + \delta_{\text{out}}\sigma_{\text{out}}^2 + \delta_{\text{out}}(1-\delta_{\text{out}})D_i} \right)^2}{\sum_i \frac{(\mu_{ci}-\mu_{c'i})^2}{((1-\delta_{\text{out}})\sigma^2 + \delta_{\text{out}}\sigma_{\text{out}}^2 + \delta_{\text{out}}(1-\delta_{\text{out}})D_i)^2}}. \tag{16}$$

Let $\lim_{N\to\infty} -8\log(B(\bar{\mu}_c, \bar{\mu}_{c'}, \Sigma, \bar{\Sigma}))$ and $\lim_{N\to\infty} -8\log(B(\hat{\mu}_c, \hat{\mu}_{c'}, \Sigma, \hat{\Sigma}))$ denote the limits of the negative logarithms of estimated error (12) of sample and MCD estimators, respectively. From (14), (15) and (16), it holds that

$$\lim_{N\to\infty} B(\hat{\mu}_c, \hat{\mu}_{c'}, \Sigma, \hat{\Sigma}) \leq \lim_{N\to\infty} B(\bar{\mu}_c, \bar{\mu}_{c'}, \Sigma, \bar{\Sigma}).$$

This implies that the limit of estimated error ratio (3) is smaller than one, i.e.,

$$\lim_{N\to\infty} \frac{B(\hat{\mu}_c, \hat{\mu}_{c'}, \Sigma, \hat{\Sigma})}{B(\bar{\mu}_c, \bar{\mu}_{c'}, \Sigma, \bar{\Sigma})} \leq 1.$$

This completes the proof of Theorem 1.

### F.4    PROOF OF LEMMA 1

In this part, we present a proof of Lemma 1. We show the almost surely convergences of sample and MCD estimators as the number of training samples $N$ goes to infinity.

**Proof of the convergence of sample estimator**. First of all, the set of training samples $\mathcal{X}_N = \{\mathbf{x}_1, \cdots, \mathbf{x}_N\}$ contains outlier samples with the fixed fraction $\delta_{\text{out}}$. So, $\mathcal{X}_N$ is from a mixture distribution $P_{\text{mix}} = (1-\delta_{\text{out}})P_{\text{data}} + \delta_{\text{out}}P_{\text{out}}$. Then mean and covariance matrix of sample estimator, $\bar{\mu}$ and $\bar{\Sigma}$, estimate mean $\mu_{\text{mix}}$ and covariance matrix $\Sigma_{\text{mix}}$ of the mixture distribution $P_{\text{mix}}$, respectively. One can induce $\mu_{\text{mix}}$ and $\Sigma_{\text{mix}}$ directly as follow:

$$\mu_{\text{mix}} = (1-\delta_{\text{out}})\mu, \quad \Sigma_{\text{mix}} = (1-\delta_{\text{out}})\sigma^2\mathbf{I} + \delta_{\text{out}}\sigma_{\text{out}}^2\mathbf{I} + \delta_{\text{out}}(1-\delta_{\text{out}})\mu\mu^T. \tag{17}$$

Since $P_{\text{mix}}$ has the finite covariance matrix, i.e., $\Sigma_{\text{mix}} < \infty$, one can apply the the Strong Law of Large Numbers[10] to the sample estimator of the mixture distribution $P_{\text{mix}}$. Then the mean and covariance matrix of sample estimator converge almost surely to the mean and covariance matrix of $P_{\text{mix}}$, respectively:

$$\bar{\mu} \overset{\text{a.s.}}{\to} \mu_{\text{mix}}, \qquad \bar{\Sigma} \overset{\text{a.s.}}{\to} \Sigma_{\text{mix}}.$$

This completes the proof of the convergence of sample estimator.

**Proof of the convergence of MCD estimator**. Consider a collection $E_q$ of subsets $\mathcal{X}_{K,q} \subset \mathcal{X}_N$ with the size $K \ (= \lfloor \delta_{\text{mcd}}N \rfloor)$, and each subset $\mathcal{X}_{K,q}$ contains the outlier samples with the fraction $q \in [0, 1]$. Then $\mathcal{X}_{K,q} \in E_q$ is from a mixture distribution $P_q = (1-q)P_{\text{data}} + qP_{\text{out}}$. One can induce that the mean $\mu_q$ and covariance matrix $\Sigma_q$ of the mixture distribution $P_q$ as (17):

$$\mu_q = (1-q)\mu, \quad \Sigma_q = (1-q)\sigma^2\mathbf{I} + q\sigma_{\text{out}}^2\mathbf{I} + q(1-q)\mu\mu^T. \tag{18}$$

---

[10]W. Feller, An Introduction to Probability Theory and Its Applications, John Wiley & Sons, 1968

Thus sample mean estimator $\bar{\mu}_{\mathcal{X}_{K,q}}$ and covariance estimator $\bar{\Sigma}_{\mathcal{X}_{K,q}}$ of a subset $\mathcal{X}_{K,q}$ converge almost surely to $\mu_q$ and $\Sigma_q$ respectively:

$$\bar{\mu}_{\mathcal{X}_{K,q}} \overset{\text{a.s.}}{\to} \mu_q, \qquad \bar{\Sigma}_{\mathcal{X}_{K,q}} \overset{\text{a.s.}}{\to} \Sigma_q,$$

by the Strong Law of Large Numbers.

On the other hand, there is a subset $\mathcal{X}_{K,q^*}^* \subset \mathcal{X}_N$ in $E_{q^*}$ which is selected by MCD estimator. Then the determinant of its covariance matrix is the minimum over all $\mathcal{X}_K \subset \mathcal{X}_N$, and $\bar{\mu}_{\mathcal{X}_{K,q^*}^*} = \widehat{\mu} \overset{\text{a.s.}}{\to} \mu_{q^*}$, $\bar{\Sigma}_{\mathcal{X}_{K,q^*}^*} = \widehat{\Sigma} \overset{\text{a.s.}}{\to} \Sigma_{q^*}$ as $N \to \infty$. Since the determinant is a continuous funciton, the Continuous Mapping Theorem[11] implies

$$\min_{\mathcal{X}_{K,q} \subset \mathcal{X}_N} \det(\bar{\Sigma}_{\mathcal{X}_{K,q}}) \overset{\text{a.s.}}{\to} \min_q \det\left(\Sigma_q\right),$$

and

$$\min_{\mathcal{X}_{K,q} \subset \mathcal{X}_N} \det(\bar{\Sigma}_{\mathcal{X}_{K,q}}) = \det(\bar{\Sigma}_{\mathcal{X}_{K,q^*}^*}) = \det(\widehat{\Sigma}) \overset{\text{a.s.}}{\to} \det\left(\Sigma_{q^*}\right).$$

Now, we'd like to show

$$\min_q \det\left(\Sigma_q\right) = \det\left(\Sigma_{q^*}\right) = \det\left(\Sigma_0\right), \tag{19}$$

to complete the proof of Lemma 1.

By the assumption $\delta_{\texttt{mcd}} \leq 1 - \delta_{\text{out}}$, $E_0$ is non-empty. It shows the existence of $\Sigma_0$. From the covariance matrix $\Sigma_q$ (18), $\det(\Sigma_q)$ is a $d$-th degree polynomial of $q$ as follow:

$$
\begin{aligned}
\det(\Sigma_q) &= \det\left((1-q)\sigma^2\mathbf{I} + q\sigma_{\text{out}}^2\mathbf{I} + q(1-q)\mu\mu^T\right) \\
&= \left((1-q)\sigma^2 + q\sigma_{\text{out}}^2\right)^{d-1}\left((1-q)\sigma^2 + q\sigma_{\text{out}}^2 + q(1-q)\mu^T\mu\right).
\end{aligned}
$$

Since the assumption gives $\sigma_{\text{out}}^2 > \sigma^2$, $\det(\Sigma_q)$ has the lower bound $\det(\Sigma_0)$ as follow:

$$
\begin{aligned}
\det(\Sigma_q) &= \left((1-q)\sigma^2 + q\sigma_{\text{out}}^2\right)^{d-1}\left((1-q)\sigma^2 + q\sigma_{\text{out}}^2 + q(1-q)\mu^T\mu\right) \\
&\geq \left(\sigma^2\right)^{d-1}\left(\sigma^2 + q(1-q)\mu^T\mu\right) \\
&\geq \left(\sigma^2\right)^{d-1}\sigma^2 = \det(\Sigma_0).
\end{aligned}
$$

Then $\det(\Sigma_q) \geq \det(\Sigma_0)$ for all $q \in [0,1]$ and the equality holds for only $q = 0$. It implies $q^* = 0$ and (19) is the shown. Therefore the mean and covariance matrix of MCD estimator converge almost surely to $\mu$ and $\sigma^2\mathbf{I}$, respectively:

$$\widehat{\mu} \overset{\text{a.s.}}{\to} \mu_0 = \mu, \quad \widehat{\Sigma} \overset{\text{a.s.}}{\to} \Sigma_0 = \sigma^2\mathbf{I}.$$

This completes the proof of Lemma 1.

---

[11]P. Billingsley, Convergence of Probability Measures, John Wiley & Sons, 1999

