# OpenReview forum: "Robust Determinantal Generative Classifier for Noisy Labels and Adversarial Attacks"
_ICLR.cc/2019/Conference_

### Official Review · AnonReviewer2 · 2018-10-26
**Interesting idea, but the paper need to improve**

**Rating:** 4
**Confidence:** 4

**Review:**

The paper proposes a new method for robustifying a pre-trained model improving its decision boundaries. The goal is to defend the model from mistakes in training labels and to be more robust to adversarial examples at test time. The main idea is to train a LDA on top of the last-layer, or many layers in its ensemble version, making use of a small set of clean labels after training the main model. Additionally, robustness to outliers is achieved by the minimum covariance determinant estimator for the LDA covariance matrix.

While I find this idea interesting and of potential practical use, I have concerns about novelty and the experimental results and overall I recommend rejection.

== Method

At a high level, the idea of imposing a mixture of gaussian structure in the feature space of a deep neural network classifier is not new. See for example [A, B]. In particular, [B] performs experiments on adversarial examples. Moreover, in spite of the authors writing that their goal is “completely different” from [Lee at al 18a, Ma et al 18a], I found the two cited papers having a similar intent and approach to the problem, but a comparison is completely missing. Without a proper comparison (formal and experimental) with these lines of work, the paper is incomplete.

Theorem 1 well supports the proposed method and it is well explained. I did not check the proofs in appendix.

Regarding the presentation, I found odd having some experimental results (page 5) before the Section on experience even have started.

== Experiments

The authors did not comment on the computational overhead of training their LDA. But I assume it is very cheap compared to training e.g. the ResNet, correct?

I also did not find an explanation of which version backward/forward losses [Patrini et al. 17] is used in the experiments: are the noise transition matrices estimated on the data or assumed to be known (for fair comparison, I would do the former).

I disagree on the importance of the numbers reported on the abstract: DenseNet on Cifar10 with 60% goes from 53.34 to 74.72. This is the improvement with the weakest possible baseline, i.e. no method to defend for noise! Looking at Table 3, which is on ResNets, I will make this point clear. Noise 60% on CIFAR10, DDGC improves 60.05-> 71.38, while (hard) bootstrap and forward do better. Even more, it seems that forward does always better than DDGC with noise 60% on every dataset. Therefore, I don’t find interesting to report how DDGC improve upon “no baseline”, because known methods do even better. Yet, it is interesting --- and I find this to be a contribution of the paper --- that DDGC can be used in combination with prior work to boost performance even further.

A missing empirical analysis is on class-conditional noise (see for example Patrini et al. 17 for a definition). An additional column on the table showing that the algorithm can also work in this case would improve the confidence that the proposed method is useful in practice. Uniform noise is the least realistic assumption for label noise.

Regarding the experiments on adversarial examples, I am not convinced of their relevance at all. There are now dozens of defence methods that work (partially) for improving robustness. I don’t think it is of any practical use to show that a new algorithm (such at DDGD) provide some defence compared to no defence. A proper baseline should have been compared.

One more unclear but important point: is Table 3 obtained by white-box attacks on the Resnet/Denset but oblivious of the MCD? Is so, I don’t think such an experiment tells the whole story: as the the MCD would arguably also be deployed for classification, the attacker would also target it.

Additionally, the authors state “we remark that accessing the parameters of the generative classifiers […] is not a mild assumption since the information about training data is required to compute them”. I don’t follow this argument: this is just part of the classifier. White box attacks are by definition performed with the knowledge of the model, what is the difference here?

Table 8 rises some concerns. I appreciate the idea of testing full white-box adversarial attacks here. But I don’t understand how it is possible that DDGC is more robust, with higher adversarial test accuracy, than in Table 3.

[A] Wen, Yandong, et al. "A discriminative feature learning approach for deep face recognition." European Conference on Computer Vision. Springer, Cham, 2016.
[B] Wan, Weitao, et al. "Rethinking feature distribution for loss functions in image classification." Proceedings of the IEEE Conference on Computer Vision and Pattern Recognition. 2018.

---

> ### Author Response · Authors · 2018-11-23
> **Responses for AnonReviewer2**
>
> We very much appreciate your valuable comments, efforts and times on our paper. Our responses for all your questions are provided below. Our major revisions in the new draft are colored by red.
>
> Q1. Comparison with [1, 2, 3, 4].
>
> The main difference between our method and [1, 2] is that we do not directly train the Gaussian mixture model, i.e., generative classifier but we post-process it on hidden feature spaces of pre-trained deep models. In addition, we study a robust inference method to handle noisy labels in training samples, while they did not. Next, [3,4] also assume clean training labels, and aim for detecting abnormal test samples after ’clean’ training. Therefore, a comparison with [1, 2, 3, 4] is not straightforward as our goal is different. We clarified this in Section 2.1 of the revised draft.
>
> Q2. Computational cost.
>
> As you expect, estimating the parameters of LDA is very cheap compared to training original deep models like ResNet and DenseNet, since it requires only one forward pass to extract the hidden features.
>
> Q3. Version of backward/forward losses.
>
> As mentioned in Appendix B of the previous draft, we use the estimated noise transition matrices for backward/forward losses. We clarified more details of experimental setups in Appendix B of the revised draft.
>
> Q4. Updated abstract and performance evaluation.
>
> As AnonReviewer 3 mentioned, our main contribution is developing a new inference method which can be used under any pre-trained deep model. In other words, our goal is not outperforming the performance of prior training methods and complementary to them, i.e., our inference method can improve the performance of any prior training methods (see our common response to all reviewers). Nevertheless, we agree with your comments that it is more meaningful to emphasize our improvement over the state-of-the-art training methods. In the abstract of the revised draft, we report our improvement over Co-teaching [5] which is the most recent and state-of-the-art training method.
>
> Q5. Evaluation on adversarial attacks.
>
> In the revised draft, we also consider optimization-based adaptive attacks against our method under the black-box setup (see Table 5) and the white-box setup (see Table 10). In both setups, our inference method is shown to be more robust compared to the softmax inference. We further show that our method further improves the robustness of deep models optimized by adversarial training (see Table 6 and 11). Such experimental results support our claim that the proposed generative classifier can improve the robustness against adversarial attacks as it utilizes multiple hidden features (i.e., harder to attack all of them). We very much appreciate your valuable comments again.
>
> [1] Wen, Y., Zhang, K., Li, Z. and Qiao, Y., A discriminative feature learning approach for deep face recognition. In ECCV, 2016.
>
> [2] Wan, W., Zhong, Y., Li, T. and Chen, J., Rethinking feature distribution for loss functions in image classification. In CVPR, 2018.
>
> [3] Lee, K., Lee, K., Lee, H. and Shin, J., A Simple Unified Framework for Detecting Out-of-Distribution Samples and Adversarial Attacks. In NIPS, 2018.
>
> [4] Ma, X., Li, B., Wang, Y., Erfani, S.M., Wijewickrema, S., Houle, M.E., Schoenebeck, G., Song, D. and Bailey, J. Characterizing adversarial subspaces using local intrinsic dimensionality. In ICLR, 2018.
>
> [5] Bo Han, Quanming Yao, Xingrui Yu, Gang Niu, Miao Xu, Weihua Hu, Ivor Tsang, and Masashi Sugiyama. Co-teaching: robust training deep neural networks with extremely noisy labels. In NIPS, 2018.
>
> Thanks a lot,
> Authors

---

> > ### Author Response · Authors · 2018-12-01
> > **After First Revision**
> >
> > Dear AnonReviewer2,
> >
> > We hope that you found our rebuttal/revision for you and other reviewers in common.
> >
> > If you have any remaining questions/concerns, please do not hesitate to let us know and we would be happy to answer.
> >
> > Thank you very much,
> > Authors

---

### Official Review · AnonReviewer3 · 2018-11-01
**Good papers but lacking of related works in deep learning with noisy labels and lacking of important baselines.**

**Rating:** 7
**Confidence:** 5

**Review:**

This paper formulates a new inference method called DDGC for noise labels and adversarial attacks. Their main idea is to induce a generative classifer on top of hidden feature spaces of the discriminative deep model. To improve the robustness, their DDGC model leverages the minimum covariance determinant (MCD) estimator. Besides, the author proposes Theorem 1 to justify their MCD-based generative classifer.

Pros:

1. The authors find a new angle for learning with noisy labels. Motivated by the fact that LDA-like generative classifer assuming the class-wise unimodal distribution might be robust, they introduce a generative classifer on top of hidden feature spaces of the discriminative deep model.

2. The authors perform numerical experiments to demonstrate the effectiveness of their framework in benchmark datasets. And their experimental result support their previous claims.

Cons:

We have two questions in the following.

1. Related works: In deep learning with noisy labels, there are three main directions, including small-loss trick [1-3], estimating noise transition matrix [4-6], and explicit and implicit regularization [7-9]. I would appreciate if the authors can survey and compare more baselines in their paper instead of listing some basic ones.

2. Experiment:
2.1 Baselines: For noisy labels, the authors should add MentorNet [1] as a baseline https://github.com/google/mentornet From my own experience, this baseline is very strong. At the same time, they should compare with VAT [7]. For adversarial attacks, the author should compare with data type from [10], and list L-FBGS [11] as a basic baseline.
2.2 Datasets: For datasets, I think the author should first compare their methods on symmetric and aysmmetric noisy data. Besides, the current paper only verifies on vision datasets. The authors are encouraged to conduct 1 NLP dataset.

References:

[1] L. Jiang, Z. Zhou, T. Leung, L. Li, and L. Fei-Fei. Mentornet: Learning data-driven curriculum for very deep neural networks on corrupted labels. In ICML, 2018.

[2] M. Ren, W. Zeng, B. Yang, and R. Urtasun. Learning to reweight examples for robust deep learning. In ICML, 2018.

[3] B. Han, Q. Yao, X. Yu, G. Niu, M. Xu, W. Hu, I. Tsang, M. Sugiyama. Co-teaching: Robust training of deep neural networks with extremely noisy labels. In NIPS, 2018.

[4] G. Patrini, A. Rozza, A. Menon, R. Nock, and L. Qu. Making deep neural networks robust to label noise: A loss correction approach. In CVPR, 2017.

[5] J. Goldberger and E. Ben-Reuven. Training deep neural-networks using a noise adaptation layer. In ICLR, 2017.

[6] S. Sukhbaatar, J. Bruna, M. Paluri, L. Bourdev, and R. Fergus. Training convolutional networks with noisy labels. In ICLR workshop, 2015.

[7] T. Miyato, S. Maeda, M. Koyama, and S. Ishii. Virtual adversarial training: A regularization method for supervised and semi-supervised learning. ICLR, 2016.

[8] A. Tarvainen and H. Valpola. Mean teachers are better role models: Weight-averaged consistency targets improve semi-supervised deep learning results. In NIPS, 2017.

[9] S. Laine and T. Aila. Temporal ensembling for semi-supervised learning. In ICLR, 2017.

[10] C. Nicholas and W. David. Towards evaluating the robustness of neural networks. In IEEE Symposium on SP, 2017.

[11] C. Szegedy, W. Zaremba, I. Sutskever, J. Bruna, D. Erhan, I. Goodfellow, and R. Fergus. Intriguing properties of neural networks. In ICLR, 2013.

---

> ### Author Response · Authors · 2018-11-23
> **Responses for AnonReviewer3**
>
> We very much appreciate your valuable comments, efforts and times on our paper. Our responses for all your questions are provided below. Our major revisions in the new draft are colored by red.
>
> Q1. More related works
>
> We updated the introduction by including more recent works [1, 2, 3, 4, 5] related to deep learning with noisy labels. In the previous draft, we only included the relevant literature which involves a single network/classifier. The updated related works utilize multiple networks, e.g., an ensemble of classifiers or meta-learning model. We also added new experimental results for them in Table 4 of the revised draft, as we mentioned in our common response to all reviewers. Thank you very much for the suggestions.
>
> Q2. Comparison with VAT [6].
>
> We remark that a targeted setting of VAT [6] is different from ours in that it is designed for improving the performance on semi-supervised learning, while our main goal is handling noisy labels in the training dataset. Due to this, we skip the comparison with VAT. Instead, as we mentioned in our common response to all reviewers, we consider more training baselines (such as MentorNet [2] and Co-teaching [3]) focusing on handling noisy labels, and show that our inference method can improve all of them.
>
> Q3. L-FBGS adversarial attacks [8].
>
> We remark that L-FBGS [8] is known to fail easily due to the near-zero gradient of loss function [7]. Instead, we consider CW attack [7] which is known to be much stronger.
>
> [1] Jacob Goldberger and Ehud Ben-Reuven. Training deep neural-networks using a noise adaptation layer. In ICLR, 2017.
>
> [2] Lu Jiang, Zhengyuan Zhou, Thomas Leung, Li-Jia Li, and Li Fei-Fei. Mentornet: Regularizing very deep neural networks on corrupted labels. In ICML, 2018.
>
> [3] Bo Han, Quanming Yao, Xingrui Yu, Gang Niu, Miao Xu, Weihua Hu, Ivor Tsang, and Masashi Sugiyama. Co-teaching: robust training deep neural networks with extremely noisy labels. In NIPS, 2018.
>
> [4] Mengye Ren, Wenyuan Zeng, Bin Yang, and Raquel Urtasun. Learning to reweight examples for robust deep learning. In ICML, 2018.
>
> [5] Eran Malach and Shai Shalev-Shwartz. Decoupling” when to update” from” how to update”. In NIPS, 2017.
>
> [6] T. Miyato, S. Maeda, M. Koyama, and S. Ishii. Virtual adversarial training: A regularization method for supervised and semi-supervised learning. ICLR, 2016.
>
> [7] C. Nicholas and W. David. Towards evaluating the robustness of neural networks. In IEEE Symposium on SP, 2017.
>
> [8] C. Szegedy, W. Zaremba, I. Sutskever, J. Bruna, D. Erhan, I. Goodfellow, and R. Fergus. Intriguing properties of neural networks. In ICLR, 2013.
>
> Thanks a lot,
> Authors

---

> > ### Comment · AnonReviewer3 · 2018-11-26
> > **"Comparison with VAT" belongs one of the main directions in deep learning with noisy labels.**
> >
> > Hi Authors,
> >
> > I appreciated your heavy revision. Please keep in mind that "VAT" is previously proposed for semi-supervised learning. However, it can be empirically used for deep learning with noisy labels.
> >
> > There have three ways to handle noisy labels. First, data perspective (Backward Correction and so on); Second, training perspective (MentorNet, Co-teaching and so on); and Lastly, regularization perspective (VAT, Mean Teacher and so on).
> >
> > We have already tested that MentorNet [1] + VAT and Co-teaching [2] + VAT will significantly boost the performance of MentorNet and Co-teaching. That is why I mention this. Due to time limits, I can understand you may not compare this baseline.
> >
> > References:
> >
> > [1] L. Jiang, Z. Zhou, T. Leung, L. Li, and L. Fei-Fei. Mentornet: Learning data-driven curriculum for very deep neural networks on corrupted labels. In ICML, 2018.
> >
> > [2] B. Han, Q. Yao, X. Yu, G. Niu, M. Xu, W. Hu, I. Tsang, M. Sugiyama. Co-teaching: Robust training of deep neural networks with extremely noisy labels. In NeurIPS, 2018.
> >
> > Regards,
> > AnonReviewer3

---

> > > ### Author Response · Authors · 2018-11-26
> > > **Response for VAT**
> > >
> > > Dear AnnoReviewer3,
> > >
> > > Thank you very much again for your clarification and suggestion.
> > >
> > > To address multiple reviewers’ concerns in common, we follow the same experimental setups of Co-teaching [1] (the most recent related work), where the authors did not consider VAT [2]. However, your suggested experiments with VAT should be very interesting, and we will add them to the final draft. As evidenced in our heavy experimental results, we strongly believe that our training-agnostic method can also improve the performance of the deep models trained with VAT, e.g., Co-teaching + VAT.
> > >
> > > Sincerely,
> > > Authors
> > >
> > > [1]  Han, B., Yao, Q., Yu, X., Niu, G., Xu, M., Hu, W., Tsang, I. and Sugiyama, M., Co-teaching: robust training deep neural networks with extremely noisy labels. In NIPS. 2018.
> > >
> > > [2] T. Miyato, S. Maeda, M. Koyama, and S. Ishii. Virtual adversarial training: A regularization method for supervised and semi-supervised learning. ICLR, 2016.

---

### Official Review · AnonReviewer1 · 2018-11-02
**Robust Determinantal Generative Classifier for Noisy Labels and Adversarial Attacks**

**Rating:** 3
**Confidence:** 4

**Review:**

Quality: A simple approach accompanied with a theoretical justification and large number of experimental results. The theoretical justification is spread out in the main body and appendices. The proof given in the appendix is overly short and not detailed enough. The large number of experiment although welcoming needs to be properly discussed and related to the state of the art numbers, including any work that the authors are referring themselves in this submission. The approach is not linked to so called Tandem approach that was/is popular in speech recognition where a generative model (GMM) is trained on top of features extracted by a neural network model.

Clarity: The simple approach is clearly described. However, the theoretical justification and experimental results are not.

Originality: The work is moderately original.

Significance: It is hard to assess given the current submission.

---

> ### Author Response · Authors · 2018-11-23
> **Responses for AnonReviewer1**
>
> We very much appreciate your valuable comments, efforts and times on our paper. Our responses for all your questions are provided below. Our major revisions in the new draft are colored by red.
>
> Q1. Updated proof.
>
> To address your concerns, we provided more detailed explanations of our proof arguments in the revised draft (see Appendix F). We also re-organized our proof completely for better understanding.
>
> Q2. Relation to Tandem approach.
>
> As you pointed out, our method is somewhat related to Tandem approach [1] in that both post-process a generative model on top of hidden features extracted by DNNs. However, the main purpose of Tandem is not for handling noisy labels. In particular, the Tandem approaches utilize the EM algorithm that should be highly influenced by outliers, while our method is specialized to be robust against them. We clarified this in Section 2 of the revised draft.
>
> [1]  Hermansky, H., Ellis, D.P. and Sharma, S., Tandem connectionist feature extraction for conventional HMM systems. In IEEE ICASSP, 2000.
>
> Thanks a lot,
> Authors

---

> > ### Author Response · Authors · 2018-12-01
> > **After First Revision**
> >
> > Dear AnonReviewer1,
> >
> > We hope that you found our rebuttal/revision for you and other reviewers in common.
> >
> > If you have any remaining questions/concerns, please do not hesitate to let us know and we would be happy to answer.
> >
> > Thank you very much,
> > Authors

---

### Author Response · Authors · 2018-11-23
**Common response for all reviewers**

We very much appreciate valuable comments, efforts, and time of the reviewers. We first address common concerns of the reviewers and other issues for each individual one separately. Revised parts in the new draft are colored by red (in particular, we updated or newly added the abstract, Section 1, 2.1 and 3, Appendix B, E, F, and Table 2, 3, 4, 5, 6, 7, 8, 10, 11, 12 and 13).

Q1. New results for comparison with more state-of-the-art training methods.

Following AnonReviewer1/3’s suggestions, we added more experimental results on other training methods including D2L [1], Co-teaching [2] and MentorNet [5] which have been achieved the state-of-the-art performance on noisy labeled datasets (see Table 3 and Table 4 of our revised draft). As expected, the new results also confirm that our inference method is training-agnostic, i.e., it can improve the performance of any prior training methods. Here, we remark that Table 3 only considers the methods training a single network (e.g., D2L [1] and Forward/Backward [3]), while those in Table 4 train multiple networks, i.e., an ensemble of classifiers (Decoupling [4] and Co-teaching [2]) or a meta-learning model (MentorNet [5]). We consider such two different setups to follow the same experimental setups of prior works [1] and [2], respectively.

Q2. New results for class-conditional (or flip) noise.

Following AnonReviewer 2/3’s suggestions, we reported the experimental results on class-conditional (called flip) noise setups of [2] (see Table 2 of our revised draft). Our method still outperforms all baseline methods by far even under such asymmetric noise setups. This confirms that our noise-agnostic method should be useful in practice.

[1] Ma, X., Wang, Y., Houle, M.E., Zhou, S., Erfani, S.M., Xia, S.T., Wijewickrema, S. and Bailey, J., Dimensionality Driven Learning with Noisy Labels. In ICML, 2018.

[2] Han, B., Yao, Q., Yu, X., Niu, G., Xu, M., Hu, W., Tsang, I. and Sugiyama, M., Co-teaching: robust training deep neural networks with extremely noisy labels. In NIPS. 2018.

[3] G. Patrini, A. Rozza, A. Menon, R. Nock, and L. Qu. Making deep neural networks robust to label noise: A loss correction approach. In CVPR, 2017.

[4] Eran Malach and Shai Shalev-Shwartz. Decoupling” when to update” from” how to update”. In NIPS, 2017.

[5] Jiang, L., Zhou, Z., Leung, T., Li, L.J. and Fei-Fei, L., MentorNet: Regularizing very deep neural networks on corrupted labels. In ICML, 2018.

Thanks a lot,
Authors

---

### Meta-Review · Area_Chair1 · 2018-12-17
**reject**

**Confidence:** 4
**Recommendation:** Reject

**Metareview:**

While the paper contains interesting ideas, the reviewers agree the experimental study can be improved.